# Nampt-mediated spindle sizing secures a post-anaphase increase in spindle speed required for extreme asymmetry

Zhe Wei [1], Jessica Greaney [1], Wei-Guo Nicholas Loh [1] & Hayden Anthony Homer [1✉]

Meiotic divisions in oocytes are extremely asymmetric and require pre- and post-anaphase-onset phases of spindle migration. The latter induces membrane protrusion that is moulded around the spindle thereby reducing cytoplasmic loss. Here, we find that depleting the NAD biosynthetic enzyme, nicotinamide phosphoribosyl-transferase (*Nampt*), in mouse oocytes results in markedly longer spindles and compromises asymmetry. By analysing spindle speed in live oocytes, we identify a striking and transient acceleration after anaphase-onset that is severely blunted following Nampt-depletion. Slow-moving midzones of elongated spindles induce cortical furrowing deep within the oocyte before protrusions can form, altogether resulting in larger oocyte fragments being cleaved off. Additionally, we find that Nampt-depletion lowers NAD and ATP levels and that reducing NAD using small molecule Nampt inhibitors also compromises asymmetry. These data show that rapid midzone displacement is critical for extreme asymmetry by delaying furrowing to enable protrusions to form and link metabolic status to asymmetric division.

[1] The Christopher Chen Oocyte Biology Research Laboratory, Centre for Clinical Research, The University of Queensland, Herston, QLD 4029, Australia.
✉email: h.homer@uq.edu.au

Mammalian oocytes undergo highly asymmetric meiotic divisions, producing a large secondary oocyte and a very small first polar body. For achieving asymmetry, an actin network displaces the spindle, and hence the spindle midzone that directs furrowing of the overlying cortex, away from the oocyte centre towards the cortex prior to anaphase[1–3]. Consequently, asymmetry is compromised if pre-anaphase-spindle migration is impaired[4–6].

A second phase of spindle migration occurring after anaphase-onset has recently been identified and is also critical for asymmetry[7]. During this second phase, the anaphase spindle migrates into the membrane thereby inducing an out-pocketing (or protrusion) that results in half of the anaphase spindle being extruded beyond the oocyte boundary[7]. Although pre-anaphase-spindle migration alone can create some measure of asymmetry, protrusion brought about by post-anaphase-onset migration is crucial for producing the smallest polar body[7]; we refer to the latter as extreme asymmetry. An outstanding question pertains to how furrowing is delayed during post-anaphase-onset spindle migration since a midzone is present for around 15–20 min prior to cleavage[7]. Movement of the midzone could be the critical event as this would prevent any single region of the overlying cortex from being exposed long enough to midzone-derived signals.

Nicotinamide adenine dinucleotide (NAD) is an essential coenzyme during redox reactions and is required by NAD-dependent enzymes such as sirtuins and poly (ADP-ribose) polymerases (PARPs)[8–10]. One of the most important routes for NAD production is the salvage pathway, the rate-limiting enzyme for which is nicotinamide phosphoribosyl transferase (Nampt). By generating the NAD precursor, nicotinamide mononucleotide (NMN) from nicotinamide (NAM), Nampt is therefore important for the wide range of functions such as oxidative phosphorylation, DNA repair and mitochondrial biogenesis that are carried out by sirtuins, PARPs and other NAD-dependent enzymes[8–10].

Here we investigate the role of *Nampt* in mouse oocytes. We find that Nampt-depletion leads to the assembly of markedly longer bipolar spindles during meiosis I (MI). These longer spindles migrate towards the cortex before anaphase and, like spindles in control oocytes, become displaced to an off-centre position by the time of anaphase-onset. Although this allows for some degree of asymmetry, polar bodies (PBs) in Nampt-depleted oocytes are nevertheless markedly larger than in controls. Unexpectedly, we find that immediately following anaphase-onset, spindle speed normally increases markedly by ~8-fold. In stark contrast, the speed of Nampt-depleted spindles increases less than 3-fold following anaphase-onset. In the absence of a post-anaphase-onset spindle acceleration, protrusion fails following Nampt-depletion, and furrowing occurs deeper within oocytes. Hence, rapid midzone motion brought about by a post-anaphase-onset spindle acceleration delays furrowing to allow time for protrusion formation that ultimately promotes extreme asymmetry. Nampt-depletion also lowers NAD and ATP levels and compromised asymmetry is replicated when NAD levels are reduced by inhibiting Nampt enzymatic activity using small molecule inhibitors. Collectively, therefore, these data link oocyte metabolic status with asymmetric division.

## Results

### Depletion of NAMPT impairs asymmetric division. 
Meiotic maturation in oocytes begins with germinal vesicle breakdown (GVBD), marking entry into M-phase of meiosis I (MI), and concludes with highly asymmetric cytokinesis during first polar body extrusion (PBE) (Fig. 1a). Following PBE, oocytes immediately enter meiosis II (MII) and arrest at metaphase II (Fig. 1a). To begin investigating *Nampt* in oocytes, we studied its endogenous expression and found that levels increased between the GV-arrested stage and MII (Fig. 1b).

To investigate Nampt function, we microinjected a *Nampt*-targeting morpholino (NamptMO) to sterically block new protein synthesis. We found that NamptMO led to a marked reduction in Nampt levels (~50%) in whole-oocyte lysates, whereas a control morpholino (ControlMO) had no effect (Fig. 1c). By 2 h following release from 3-isobutyl-1-methylxanthine (IBMX; a phospho-diesterase inhibitor that sustains GV-arrest), Nampt-depleted and mock-depleted oocytes underwent GVBD at comparable rates (~80%; Fig. 1d). Using time-lapse imaging, we found that Nampt-depleted oocytes also completed MI marked by PBE at similar rates to uninjected and mock-depleted oocytes (~80%; Fig. 1e).

Strikingly, however, we found that almost 45% of Nampt-depleted oocytes exhibited markedly enlarged PBs (Fig. 1f–h; Supplementary Movie 1). This was specific to Nampt-depletion as co-injection of recombinant protein to restore Nampt to endogenous levels (Supplementary Fig. 1) significantly reduced the proportion of NamptMO-injected oocytes with enlarged PBs as well as mean PB sizes (Fig. 1f–h). Nampt-depletion also appeared to impair asymmetry during the second meiotic division as there was an increased proportion of enlarged second PBs following activation of MII-arrested Nampt-depleted oocytes (Supplementary Fig. 2). Thus, Nampt-depletion did not affect the ability to resume or complete meiotic maturation but severely compromised division asymmetry.

### Nampt-depleted spindles migrate normally before anaphase. 
Deranged asymmetry is often linked with severe defects in spindle migration to the cortex prior to anaphase-onset[5,6,11]. We therefore undertook an in-depth analysis of spindle migration using time-lapse imaging of live oocytes expressing fluorescent histone H2B and SiR-tubulin for labelling chromosomes and spindles, respectively[7,12].

We found that very similar to controls, Nampt-depleted oocytes assembled bipolar spindles between 2.5 and 5 h post-GVBD that were located close to the oocyte centre in ~90% of cases (Fig. 2a–c; Supplementary Movies 2 and 3). In both control and Nampt-depleted oocytes, spindles then migrated so that 77–90% were located at, or close to, the cortex by the time of anaphase-onset (Fig. 2a, d; Supplementary Movies 2 and 3). To further investigate spindle migration, we measured the leading pole-to-cortex and spindle centre-to-cortex distances (Fig. 2e) at hourly intervals and found that both distances declined at similar rates in the lead up to anaphase in control and Nampt-depleted oocytes (Fig. 2f, g). Thus, impaired asymmetry following Nampt-knockdown was not due to failed pre-anaphase-spindle migration.

### Nampt-depletion results in longer bipolar spindles. 
Next, we analysed spindle dimensions and anaphase-spindle behaviour. We found that the mean length of bipolar spindles following Nampt-depletion (39.9 ± 1.3 μm) was significantly longer than in controls (33.6 ± 1.9 μm; $P = 0.008$; Fig. 3a, b). During anaphase, control spindles elongated by ~7.2 μm, representing a 24% increase over pre-anaphase-spindle length, similar to the ~8.0 μm and 23.0% increase seen after Nampt-knockdown (Fig. 3a–d). Unlike spindle length, spindle width was not affected following Nampt-depletion and exhibited similar dimensions to controls when bipolar spindles first formed (Fig. 3e) and throughout anaphase (Fig. 3f).

The foregoing suggested that Nampt was required for restraining spindle length. We therefore asked whether over-

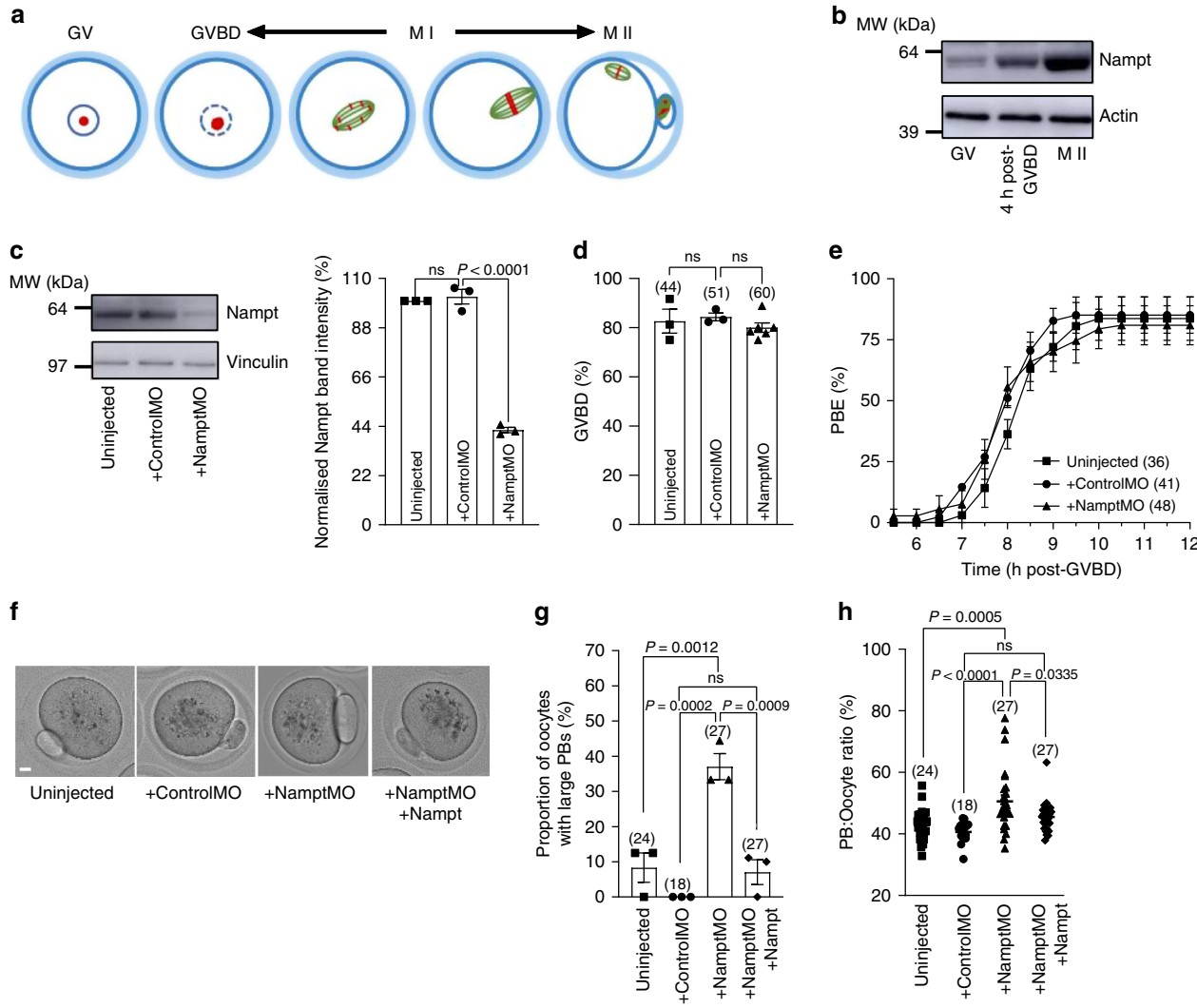

**Fig. 1 Depletion of Nampt compromises asymmetry. a** Schematic depicting stages of meiotic maturation and asymmetric division in oocytes. **b** Western blot of endogenous Nampt levels during meiotic maturation; $n = 30$ oocytes per lane. Representative blot from three independent experiments is shown. **c** Uninjected, controlMO- and NamptMO-injected oocytes were immunoblotted for Nampt. Vinculin served as a loading control (left); $n = 15$ oocytes per lane. Experiments were repeated three times. Quantification of Nampt protein levels (right). **d, e** GVBD rates 2 h following release from IBMX (**d**) and PBE rates relative to time of GVBD (**e**) in uninjected, controlMO- and NamptMO-injected oocytes. **f** Brightfield images of representative uninjected, mock-depleted and Nampt-depleted oocytes as well as Nampt-depleted oocytes co-injected with recombinant Nampt. Scale bar, 10 μm. Representative images from three independent experiments are shown. **g** Proportion of oocytes with large PBs. See Methods for further details. **h** Ratio of PB width to oocyte diameter. Oocyte numbers are shown in parenthesis from three independent experiments (**d, e, g, h**). Data in graphs are presented as the mean ± SEM. Data points are shown in **c, d** and **g**. *P*-values are shown in the graphs, ns denoted $P > 0.05$. Statistical comparisons were made using one-way ANOVA in **c, d, g** and **h**. See also Supplementary Movie 1. Source data are provided as a Source Data file.

expressing Nampt might produce shorter spindles. We tested this by microinjecting recombinant Nampt protein (Supplementary Fig. 3a). Entirely consistent with a requirement for sizing spindles, three to four times as many Nampt over-expressing oocytes exhibited short spindles compared with controls (Supplementary Fig. 3b, c).

Thus, Nampt regulates spindle length in oocytes. Importantly, Nampt-depletion resulted in markedly longer pre-anaphase bipolar spindles. Anaphase spindles were also longer following Nampt-depletion, but this was a consequence of longer pre-anaphase spindles rather than excessive elongation.

**Nampt-depletion compromises anaphase-spindle acceleration.**
Control spindles undergo migration following anaphase-onset that is critical for protrusion formation and extreme asymmetry[7]. Nampt-

depletion did not cause an inherent defect that abolished migration as spindles underwent pre-anaphase migration (Fig. 2f, g). By measuring lagging pole-to-cortex (Fig. 4a) and midzone-to-cortex (Fig. 4b) distances at 5-min intervals during anaphase-progression, we found that Nampt-depleted spindles also migrated after anaphase-onset (Fig. 4c, d). However, when compared to anaphase spindles in control oocytes, Nampt-depleted spindles moved significantly shorter distances over the same time intervals (Fig. 4c, d). This suggested that although Nampt-depleted spindles moved, their speed was impaired during the post-anaphase-onset period.

We noticed that in contrast to migration during the pre-anaphase period in controls (Fig. 2f, g), the distance spindles travelled during the first 5 min after anaphase-onset seemed markedly less than the distance covered over the second 5 min (Fig. 4c, d), suggesting that spindle speed was not uniform during anaphase. To specifically study how spindle speed changed during

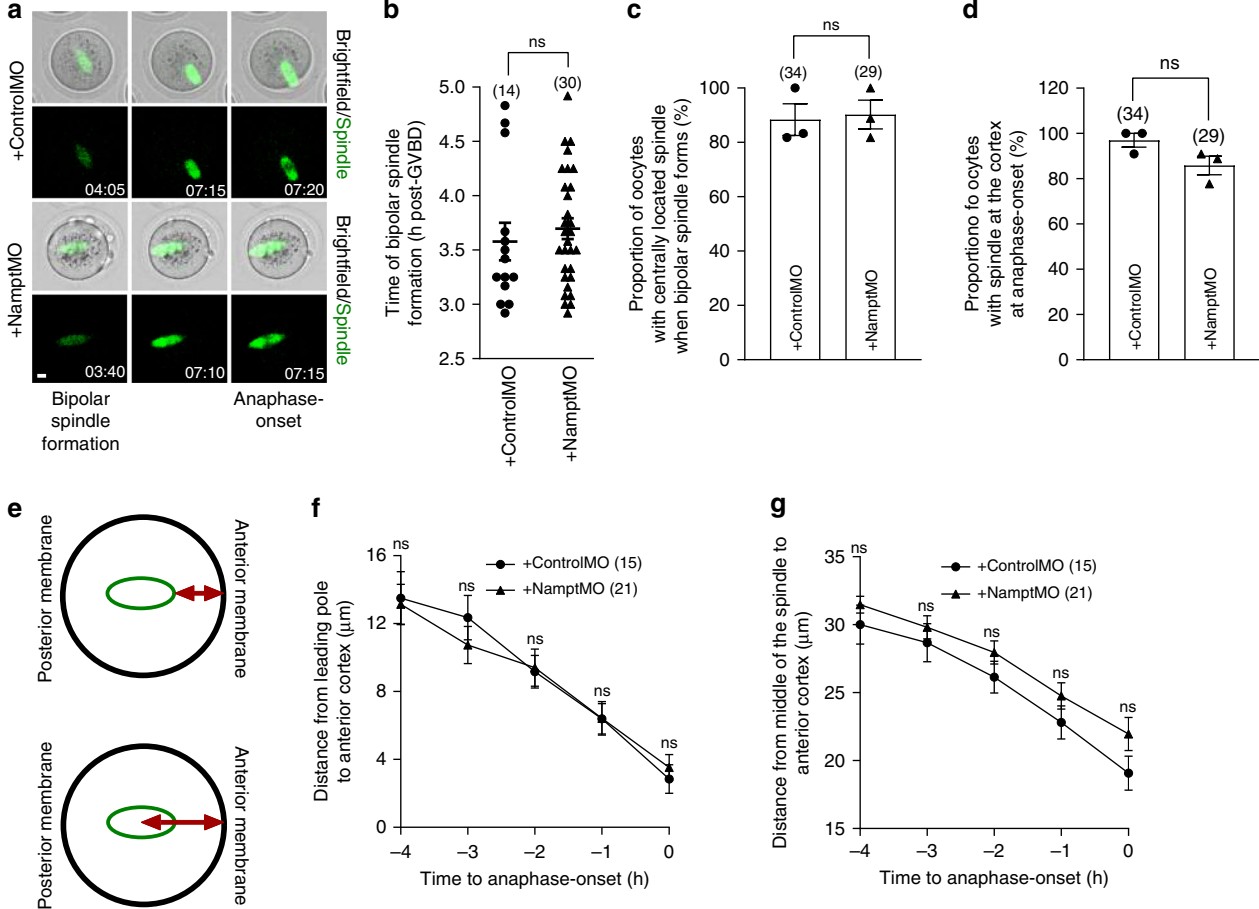

**Fig. 2 Intact pre-anaphase-spindle migration following Nampt-depletion. a** Shown are panels comprised of selected brightfield and fluorescence frames from representative time-lapse series of oocytes with SiR-tubulin-labelled spindles. Note that in both ControlMO- and NamptMO-injected oocytes, the bipolar spindle is located close to the oocyte centre when first formed and then migrates to the cortex prior to anaphase-onset. Times are shown as h: min relative to GVBD. Scale bar, 10 μm. Representative images from three independent experiments are shown. **b** Timing of bipolar spindle formation. **c** Proportion of oocytes with spindles located in the centre at the time of bipolar spindle formation. **d** Proportion of oocytes with spindles at the cortex when anaphase occurred. **e** Schematics depicting the measurements that were plotted in **f** and **g**. **f** Changes in leading pole-to-cortex distance over time. **g** Changes in spindle centre-to-cortex distance over time. Oocyte numbers are shown in parenthesis from three independent experiments (**b–d**, **f** and **g**). Data in graphs are presented as the mean ± SEM. Data points are shown in **c** and **d**. *P*-values, ns denoted *P* > 0.05. Statistical comparisons were made using either two-tailed Student's *t*-test in **b–d** or two-way repeated measures ANOVA in **f** and **g**. See also Supplementary Movies 2 and 3. Source data are provided as a Source Data file.

the peri-anaphase period, we used 3D surface tracking in Imaris. We found that spindle speed prior to anaphase was relatively constant (Fig. 4e). Strikingly, however, shortly after anaphase-onset, spindle speed in controls increased by 8-fold above pre-anaphase speeds and this occurred over a short interval of ~10 min (Fig. 4e, f, h; Supplementary Movie 4). We then analysed Nampt-depleted spindles and found that their pre-anaphase speeds were very similar to controls (Fig. 4e). Highly significantly, however, spindle speed following anaphase-onset after Nampt-depletion increased only marginally (<3-fold) above pre-anaphase speeds (Fig. 4e, g, h; Supplementary Movie 5).

Thus, in controls, spindles exhibit a marked increase in speed over a short period following anaphase-onset. In stark contrast, longer Nampt-depleted spindles completely lack this acceleration. Altogether, these data suggest that Nampt-mediated restraint of spindle size and/or another function of Nampt could be critical for enabling rapid movement of the anaphase spindle and hence the midzone.

**Protrusion formation requires anaphase-spindle acceleration.** We hypothesised that the rapid midzone movement we identified

here would be critical for delaying furrowing thereby enabling protrusions to form prior to cleavage. If correct, one prediction is that in the absence of rapid midzone movement, protrusion formation would be compromised.

We quantified key facets of protrusion-mediated division, which occurred in >90% of controls (Fig. 5a, b). First, we found that maximal protrusion widths exhibited little variation and were always less than double the width of anaphase spindles (Fig. 5c). Second, both sides of the membrane at the base of the protrusion typically ingressed at a position that very closely coincided with the geometrical middle of the spindle midzone (Fig. 5a, d; Supplementary Movie 6)[7]. Third, and entirely consistent with furrowing occurring halfway along the spindle, the maximal PB height at the time of furrowing correlated very closely with, and was directly proportional to, spindle length (Fig. 5e, f). These data enable us to define protrusions as membrane out-pocketings whose maximal dimensions are <2-fold the anaphase-spindle width and roughly half the anaphase-spindle length.

In contrast to controls, we found that about half of Nampt-depleted oocytes did not exhibit distinct protrusions (Fig. 5b) but instead produced what appeared to be membrane bulges (Fig. 5g;

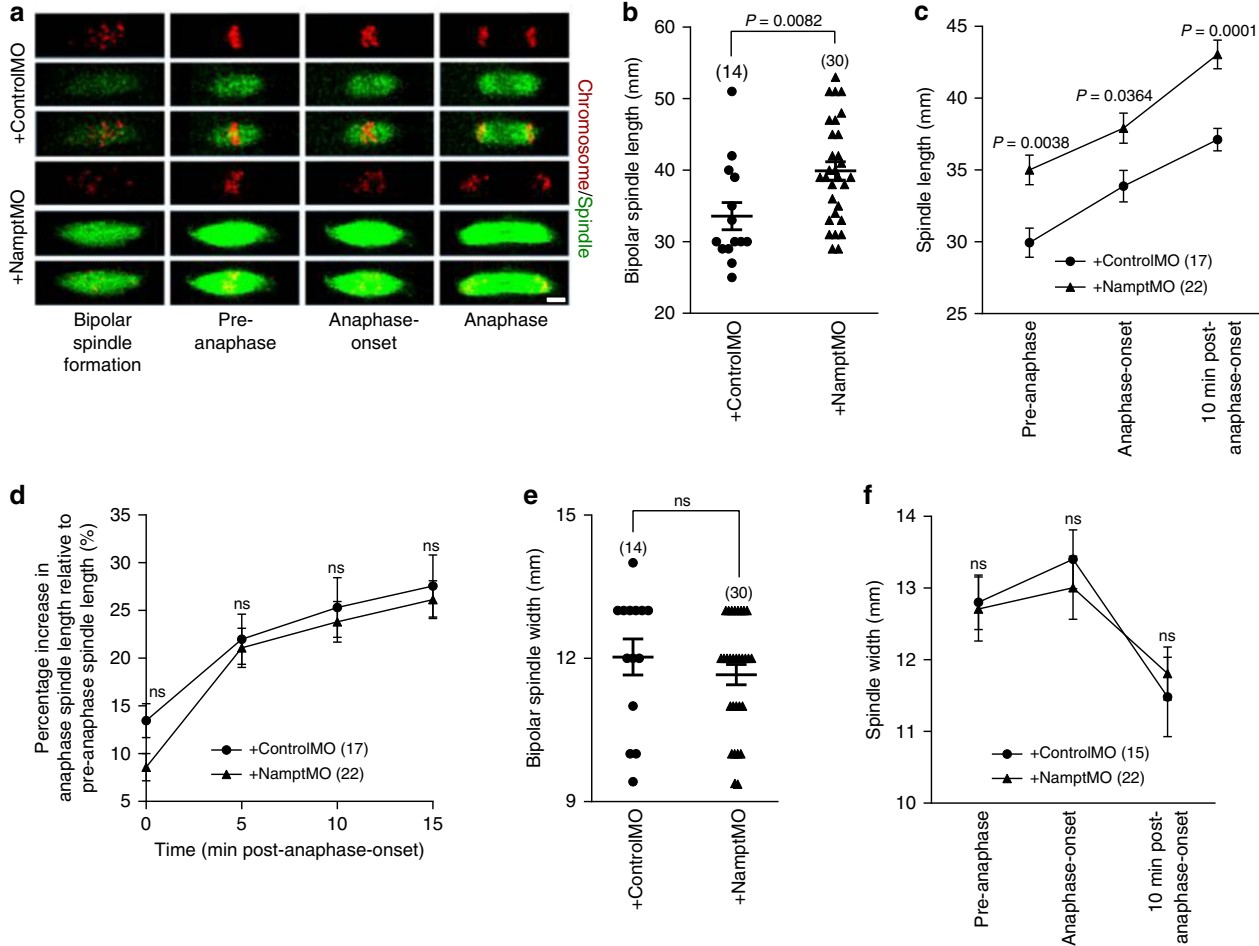

**Fig. 3 Nampt-depletion increases spindle length but not anaphase-spindle elongation. a** Shown are panels comprised of selected frames from representative time-lapse series of oocytes with SiR-tubulin-labelled spindles and expressing H2B-RFP to label chromosomes. Scale bar, 10 μm. Representative images from three independent experiments are shown. **b** Spindle length at the time when bipolar spindles first formed. **c** Measurements of spindle lengths at 5 min prior to anaphase-onset (pre-anaphase), anaphase-onset and 10 min post-anaphase-onset. **d** Percentage increase in spindle length during anaphase relative to pre-anaphase-spindle length. **e** Spindle width at the time when bipolar spindles first formed. **f** Spindle width at pre-anaphase, anaphase-onset and 10 min post-anaphase-onset. Oocyte numbers are shown in parenthesis from three independent experiments (**b**–**e**, **f**). Data in graphs are presented as the mean ± SEM. P-values are shown in the graphs, ns denoted $P > 0.05$. Statistical comparisons were made using either two-tailed Student's t-test in **b** and **e** or two-way repeated measures ANOVA in **c**, **d** and **f**. Source data are provided as a Source Data file.

Supplementary Movie 7). Strikingly, the widths of bulges were markedly larger than protrusions, ranging from 2.5 to 5 times the width of anaphase spindles (Fig. 5c). Furthermore, unlike controls in which bilateral furrowing occurred at the base of the protrusion roughly halfway along the anaphase spindle (Fig. 5a, d), in Nampt-depleted oocytes, the membrane deep within the oocyte ingressed unilaterally, on the side that was closer to the midzone (Fig. 5g; Supplementary Movie 7). In addition, this membrane ingression occurred more randomly along the spindle length, typically in an off-centre position closer to the leading spindle pole (Fig. 5d, g; Supplementary Movie 7). Consequently, for this non-protrusion pathway, the height of the PB exhibited little or no correlation with spindle length (Fig. 5h).

Thus, during normal PBE, the protrusion is closely moulded around the rapidly advancing anaphase spindle and furrowing occurs at the base of the out-pocketing thereby reducing the cytoplasmic volume lost at division (Fig. 5a). Conversely, slow-moving elongated Nampt-depleted spindles are unable to induce protrusions before furrowing begins deep within the oocyte resulting in enlarged PBs; rather than being a true membrane out-pocketing induced by the spindle, localised cortical narrowing created by membrane ingression gives the appearance of bulging

beyond the narrowed region (Fig. 5g). Under these conditions, PB size is not directly related to spindle length as it is not the elongated spindle per se that leads to enlarged PBs, rather it is compromised spindle movement. We therefore conclude that rapid anaphase-spindle motion is critical for a protrusion-based pathway that limits cytoplasmic loss to the PB.

**Nampt-depletion induces metabolic derangements**. As Nampt acts via the salvage pathway to sustain cellular NAD[8], we next asked whether Nampt-depletion impacted NAD levels. We measured NAD levels in oocytes as before[13] and found that they were significantly reduced following Nampt-depletion (Fig. 6a) suggesting that reduced NAD availability might compromise asymmetry. To investigate this further, we treated oocytes with the highly specific small molecule inhibitor of enzymatic Nampt activity, FK866[14,15]. At a concentration of 1 μM, FK866 reduced NAD levels by ~40% and resulted in enlarged PBs in around one-third of oocytes (Fig. 6b, c). Treatment with another Nampt inhibitor, STF-118804[16], induced a similar reduction in NAD and, importantly, also resulted in enlarged PBs in a similar proportion of cases (Fig. 6b, d). In contrast, in the presence of higher NAD levels in DMSO-treated controls, enlarged PBs were rarely

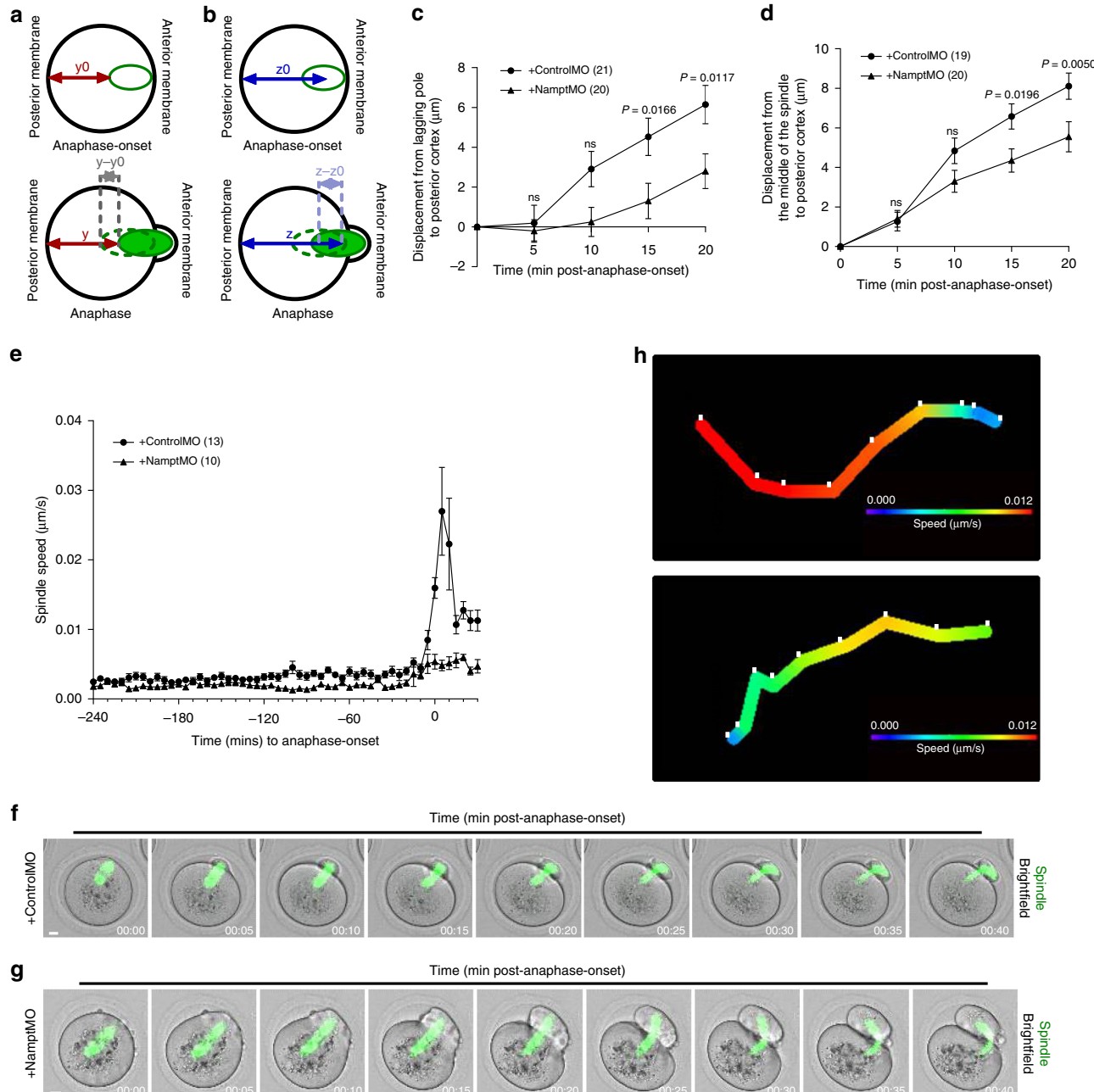

**Fig. 4 Post-anaphase-onset spindle acceleration. a, b** Schematics depicting the spindle displacement measurements plotted in **c** and **d**. **c, d** Posterior pole displacement ($y - y^0$ (**a**)) (**c**) and midzone displacement ($z - z^0$; (**b**)) (**d**) during anaphase. **e** Mean spindle migration speeds relative to anaphase-onset for mock-depleted and Nampt-depleted oocytes. Note the increase in speed in mock-depleted oocytes that is absent after Nampt-depletion. **f, g** Shown are panels comprise selected brightfield and fluorescence frames from representative time-lapse series of mock-depleted (**f**) and Nampt-depleted (**g**) oocytes. Time, h:min relative to anaphase-onset. Scale bars, 10 μm. Representative images from three independent experiments are shown. **h** Tracks of spindle migration speeds during anaphase for the oocytes shown in **f** and **g**. White marks correspond to the position of the spindle at each consecutive time point shown in **f** and **g**. Note that in the mock-depleted oocyte shown, spindle speed is maximal (red track) between anaphase-onset and 10-min post-anaphase-onset. Note that the distance between white tick marks does not correspond to the distance moved by spindles as this track is a 2D representation of motion that occurred in three dimensions. Oocyte numbers are shown in parenthesis from three independent experiments (**c–e**). Data in graphs are presented as the mean ± SEM. P-values are shown in the graphs, ns denoted $P > 0.05$. Statistical comparisons were made using two-way repeated measures ANOVA in **c** and **d**. See also Supplementary Movies 4 and 5. Source data are provided as a Source Data file.

observed (<5%) (Fig. 6b–d). We reasoned that if reduced NAD leads to impaired asymmetry following Nampt-depletion as the foregoing suggest, then bypassing the requirement for Nampt within the NAD salvage pathway should restore asymmetry. This was indeed the case as co-injection of NMN—the product of the Nampt enzymatic reaction within the salvage pathway[8]—reduced

spindle lengths as well as the proportion of Nampt-depleted oocytes with enlarged PBs (Fig. 6e, f). Thus, Nampt-depletion reduces NAD levels, which leads to compromised asymmetry.

Reduced NAD could impact spindles by impairing proteins directly involved in spindle assembly. One member of the NAD-dependent sirtuin family of deacetylases, Sirt2, is known to

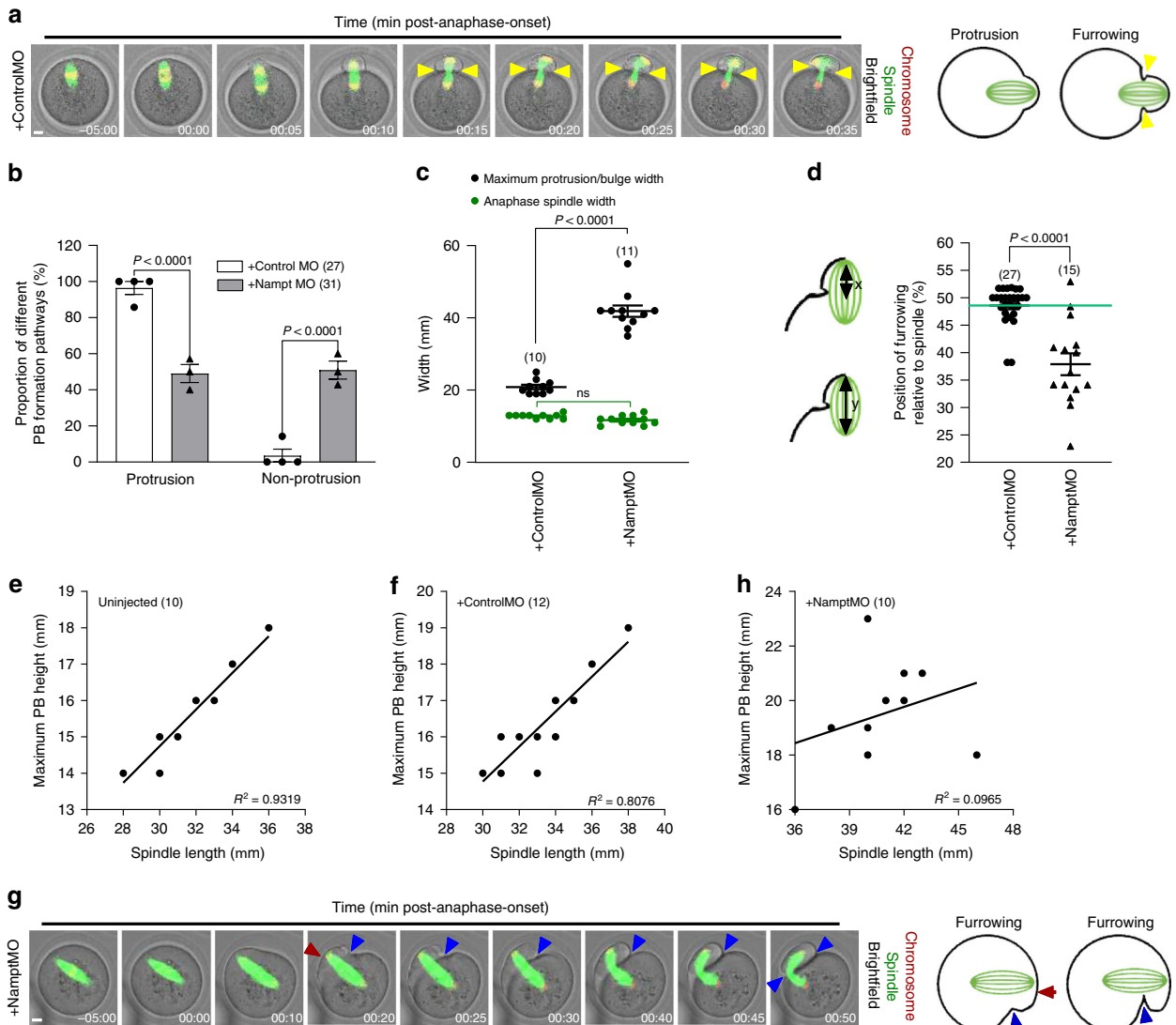

**Fig. 5 Characteristics of protrusion and non-protrusion pathways. a** Shown is a panel comprised of selected frames from representative time-lapse series of mock-depleted oocytes with SiR-tubulin-labelled spindles and expressing H2B-RFP to label chromosomes. Yellow arrowheads highlight the position of symmetrical furrowing occurring at the base of the protrusion in a control. Shown to the right is a schematic of the protrusion pathway. Scale bar, 10 μm. Representative images from three independent experiments are shown. **b** Proportions of mock- and Nampt-depleted oocytes which undergo protrusion- and non-protrusion pathways for PB formation. **c** Spindle widths and maximum protrusion/bulge widths. **d** Position of furrowing along the length of the spindle. The distance between the leading pole and the position along the spindle that is abutted by the ingressing furrow ($x$) was expressed as a percentage of the total spindle length ($y$) in mock-depleted and Nampt-depleted oocytes as illustrated in the schematic. Green line, highlights the mean value for mock-depleted oocytes. **e–h** Correlation between maximum PB height and spindle length in uninjected (**e**), mock-depleted (**f**) and Nampt-depleted oocytes (**h**). **g** Shown is a panel comprised of selected frames from representative time-lapse series of Nampt-depleted oocytes with SiR-tubulin-labelled spindles and expressing H2B-RFP to label chromosomes. Blue arrowhead highlights unilateral furrowing in the absence of a clear protrusion. Red arrowhead highlights the bulge. Shown to the right is a schematic of the non-protrusion pathway. Scale bar, 10 μm. Representative images from three independent experiments are shown. Times are h:min relative to anaphase-onset. Oocyte numbers are shown in parenthesis from three independent experiments (**b–f**, **h**). Data in graphs are presented as the mean ± SEM. Data points are shown in **b**. $P$-values are shown in the graphs, ns denoted $P > 0.05$. Statistical comparisons were made using either two-way ANOVA in **b** and **c** or a two-tailed Student's $t$-test in **d**. See also Supplementary Movies 6 and 7. Source data are provided as a Source Data file.

deacetylate tubulin[17]. Consistent with compromised Sirt2 activity, we found that levels of acetylated α-tubulin were increased following Nampt-depletion and reduced when Nampt was over-expressed (Supplementary Fig. 4a). If Sirt2 disruption was responsible for spindle defects, then compromising Sirt2 function should also derail asymmetry. To test this, we microinjected the catalytically inactive Sirt2-mutant construct, Sirt2-H187Y,[18] in oocytes. Notably, however, we did not observe any increase in either spindle length or PB size with Sirt2-H187Y (Supplementary

Fig. 4b, c). We also depleted Sirt2 using a previously characterised *Sirt2*-targeting morpholino[18] and again did not find any increase in PB size (Supplementary Fig. 4d, e). Reduced NAD could also impact NAD-dependent processes such as mitochondrial activity that are indirectly required for spindle function. Significantly, we found that oocyte ATP levels were markedly reduced following Nampt-depletion or Nampt-inhibition (Fig. 6g, h). Thus, our data raise the possibility that metabolic defects could be a major contributor to defective spindle function following Nampt-depletion.

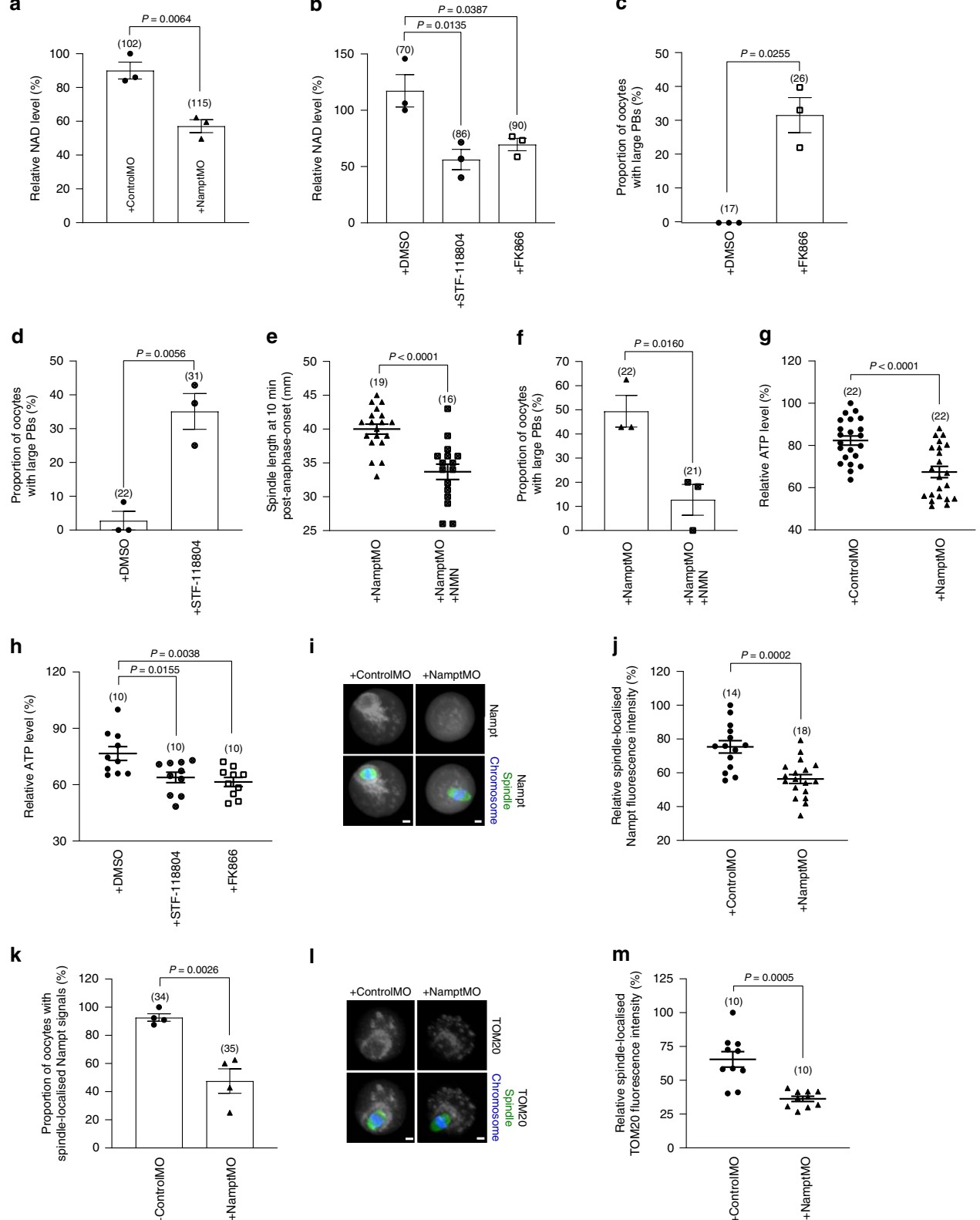

We were curious as to why spindles would be especially vulnerable to metabolic derangements following Nampt-depletion. Oocytes are deficient in glycolysis and are therefore heavily reliant on mitochondrial oxidative phosphorylation for generating ATP[19]. To support the high energetic demands of spindle assembly and migration, which occur over several hours, at least 40% of all mitochondria in oocytes congregate around the spindle[20]. Strikingly, we found that Nampt co-localised strongly with mitochondria around the periphery of bipolar spindles in >85% of control oocytes (Fig. 6i–l) suggesting that this pool of Nampt could reside within mitochondria[21]. Highly significantly, both peri-spindle Nampt and mitochondrial enrichment were severely diminished following Nampt-knockdown (Fig. 6i–m). Thus, Nampt localises strongly with mitochondria to the spindle

**Fig. 6 Nampt co-localises with peri-spindle mitochondria and asymmetry is associated with reduced NAD and ATP levels. a, b** Relative NAD levels in mock-depleted and Nampt-depleted oocytes (**a**) and in DMSO-, STF-118804- and FK866-treated oocytes (**b**). **c, d** Proportion of oocytes with large PBs in DMSO-treated oocytes versus either FK866-treated (**c**) or STF-118804-treated (**d**) oocytes. **e** Spindle length at 10 min post-anaphase-onset in NamptMO-injected and NamptMO+NMN-injected oocytes. **f** Proportion of oocytes with large PBs in NamptMO-injected and NamptMO+NMN-injected oocytes. **g** Relative ATP levels in mock-depleted and Nampt-depleted oocytes. **h** Relative ATP levels in DMSO-, STF-118804- and FK866-treated oocytes. **i** Representative images of mock-depleted and Nampt-depleted oocytes labelled with anti-Nampt antibody (monochrome), anti-β-Tubulin antibody for labelling the spindle (green) and Hochest for staining chromosomes (blue). Representative images from three independent experiments are shown. **j** Quantification of spindle-localised Nampt levels in mock-depleted and Nampt-depleted oocytes. **k** Proportions of mock- and Nampt-depleted oocytes with spindle-localised Nampt signals. **l** Representative images of mock-depleted and Nampt-depleted oocytes labelled with anti-TOM20 antibody (monochrome), anti-β-Tubulin antibody (green) and Hochest for chromosomes (blue). Representative images from three independent experiments are shown. **m** Quantification of spindle-localised TOM20 levels in mock-depleted and Nampt-depleted oocytes. Scale bars, 10 μm. Oocyte numbers are shown in parenthesis from three independent experiments (**a–h**, **j**, **k** and **m**). Data in graphs are presented as the mean ± SEM. Data points are shown in **a–d**, **f** and **k**. P-values are shown in the graphs. Statistical comparisons were made using either a one-way ANOVA in **b** and **h** or two-tailed Student's t-test in **a**, **c–g**, **j**, **k** and **m**. Source data are provided as a Source Data file.

perimeter and both are compromised by Nampt-depletion. As mitochondrial Nampt has been shown to be critical for mitochondrial metabolism in some studies[21], these data collectively suggest that Nampt-depletion disrupts a pivotal peri-spindle Nampt-mitochondria metabolic node key for supporting normal spindle activities.

**Changes in F-actin following Nampt-depletion.** The foregoing showed that either depletion or inhibition of Nampt reduced the levels of both NAD and ATP in oocytes. Due to the key role of the actin cytoskeleton in asymmetric division and the importance of ATP and Nampt-dependent NAD for actin regulation[22–24], we therefore investigated whether altered F-actin behaviour might contribute to defective asymmetry following Nampt-depletion. Cytoplasmic F-actin polymerisation undergoes a marked increase at the time of anaphase-onset in oocytes detectable using a fluorescent utrophin probe (UtrCH-mCherry)[2,7,25,26]. As this increase provides a propulsive force to the spindle, we asked whether Nampt-depletion may have affected its magnitude. We found that UtrCH-mCherry fluorescence rose sharply within minutes of anaphase-onset in mock-depleted oocytes and remained elevated for over 30 min (Supplementary Fig. 5a) concurrently with the period of accelerated spindle motion. Importantly, UtrCH-mCherry fluorescence increased to similar levels, and for a similar duration, in Nampt-depleted oocytes (Supplementary Fig. 5a).

UtrCH-mCherry also highlights localised cortical actin thickening brought about by encroachment of the leading spindle pole during late MI. We measured the relative thickness of the UtrCH-mCherry signal in the vicinity of the leading spindle pole immediately prior to anaphase-onset and found no difference between mock- and Nampt-depleted oocytes (Supplementary Fig. 5b, c).

Thus, there were no overt differences in either cytoplasmic or cortical F-actin behaviour during the peri-anaphase period following Nampt-depletion.

**Nampt influences spindle size as part of the Mos pathway.** The Mos/MAPK pathway is a well-known determinant of spindle size in oocytes with loss of Mos causing enlarged spindles[7,27]. Notably, we observed several striking similarities between Nampt-depleted oocytes and oocytes depleted of Mos using a validated morpholino[7,28,29]. First, we found that the mean length of bipolar spindles in Mos-depleted oocytes was significantly longer than in controls (Fig. 7a). Second, and as reported before[7,27], we found that Mos-depletion increased anaphase-spindle lengths as well as PB size (Fig. 7b–d). Third, we found that Mos-depleted spindles exhibited a severely compromised post-anaphase-onset increase in speed (Fig. 7e). Finally, Mos-depletion led to unilateral

furrowing positioned deep within the oocyte in over one-third of cases versus none in mock-depleted oocytes (Fig. 7f).

Interestingly, inhibition of MAPK reduces Nampt levels in melanoma cells[30] raising the possibility that in oocytes, Mos/MAPK impacts spindle size by regulating Nampt thereby explaining the strong similarities between the Mos- and Nampt-depletion phenotypes. Entirely consistent with this, we found that Nampt levels were markedly reduced in Mos-depleted oocytes (Fig. 7g). Furthermore, co-expressing recombinant Nampt in Mos-depleted oocytes significantly reduced both spindle length and PB size (Fig. 7b–d). Collectively therefore, these data strongly support that Nampt is part of the Mos/MAPK pathway for regulating spindle length and asymmetry in oocytes.

## Discussion
Thus far, spindle migration has been considered in binary terms as either being present or absent and it was unknown whether any specific characteristics of spindle motility are necessary for asymmetry. Our findings identify a unique property of spindle motion—an increase in speed—occurring specifically after anaphase-onset. We argue that this rapid acceleration is critical to delay furrowing and facilitate a protrusion-based pathway; this pathway moulds the smallest out-pocketing of cytoplasm around the leading spindle pole and is therefore the most economical mechanism for restricting the volume of cytoplasm lost. We make the distinction here between asymmetric division, which produces one daughter-cell that is larger than the next, and extreme asymmetric division, which generates the smallest PBs. Pre-anaphase migration to the cortex is sufficient for the former whereas the latter requires post-anaphase-onset spindle migration and protrusion.

Following Nampt-depletion, asymmetric division is still achieved, however, extreme asymmetry is compromised. Our findings indicate that this is not simply because longer spindles position the midzone, and hence furrowing, deeper within the oocyte, since ordinarily, furrowing does not occur within the oocyte but at the protrusion base that coincides with the oocyte surface[7]. Instead, our results support that an entirely different non-protrusion pathway occurs. Indeed, furrowing in Nampt-depleted oocytes often occurs closer to the leading spindle pole than the geometric spindle midline, yet very large PBs still result. Under these circumstances, furrowing occurs through the oocyte itself rather than at the base of protrusions that tightly mould around spindles.

We find that Nampt is critical for anaphase-spindle acceleration by furnishing NAD via the salvage pathway. NAD is an essential co-factor for NAD-dependent enzymes such as sirtuins[10]. Although one of the sirtuins, Sirt2, regulates tubulin acetylation status[17,31], and seemed an obvious candidate for modulating spindle behaviour, we did not find that Sirt2

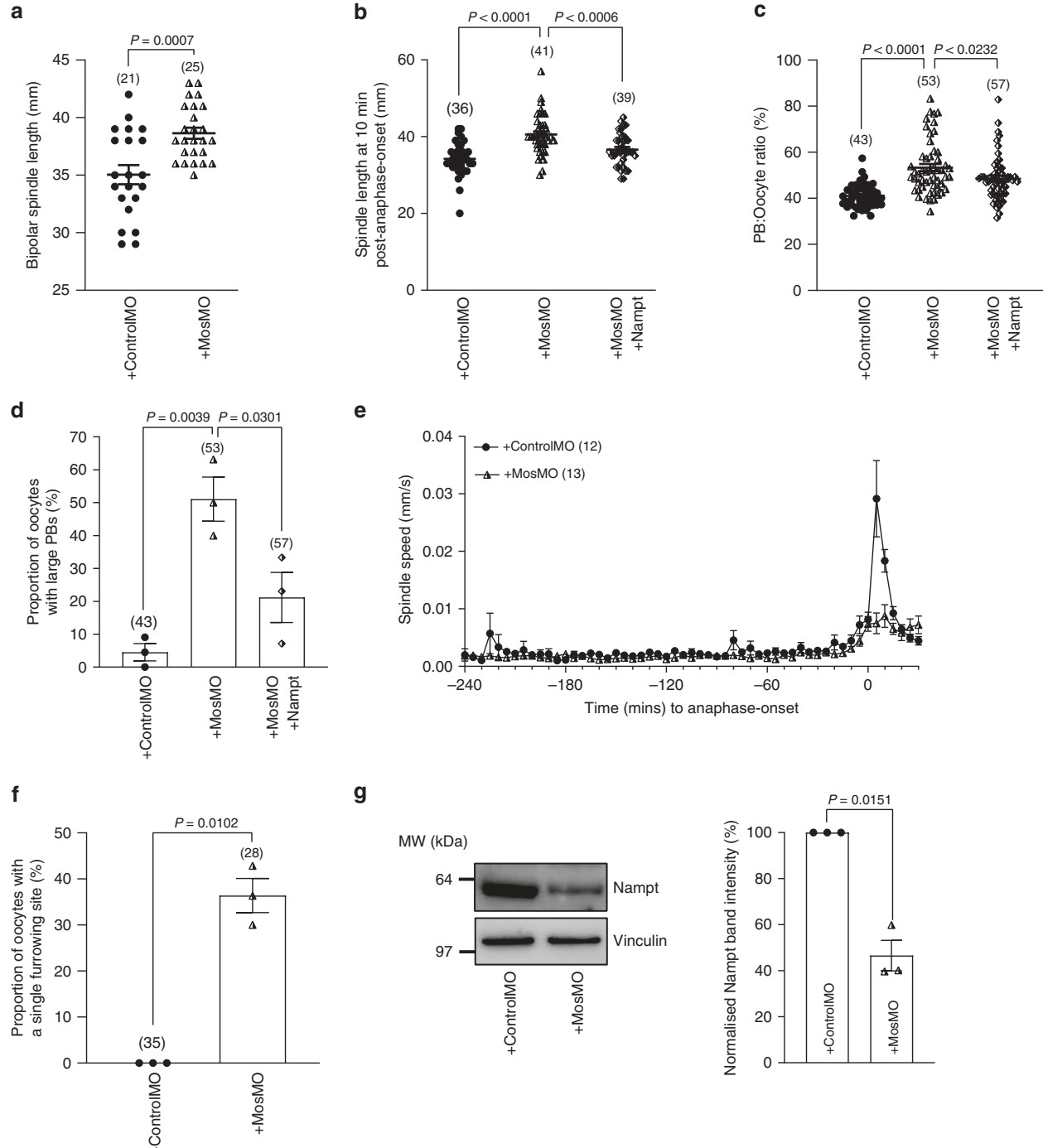

**Fig. 7 A Mos–Nampt pathway for asymmetry. a, b** Spindle lengths at the time when bipolar spindles first formed (**a**) and 10 min post-anaphase-onset (**b**). **c** Ratio of PB width to oocyte diameter. **d** Proportion of oocytes with large PBs. See Methods for further details. **e** Mean spindle migration speeds relative to anaphase-onset for mock-depleted and Mos-depleted oocytes. **f** Proportion of oocytes with a single furrowing site. **g** ControlMO- and MosMO-injected oocytes were immunoblotted for Nampt. Vinculin served as a loading control (left); $n = 15$ oocytes per lane. Experiments were repeated three times. Quantification of Nampt protein levels (right). Note that Nampt levels were markedly reduced following Mos-depletion. Oocyte numbers are shown in parenthesis from three independent experiments (**a–f**). Data in graphs are presented as the mean ± SEM. Data points are shown in **d**, **f** and **g**. P-values are shown in the graphs. Statistical comparisons were made using either two-tailed Student's t-test in **a**, **f** and **g** or one-way ANOVA in **b–d**. Source data are provided as a Source Data file.

disruption impacted asymmetry. We recently found that loss of Sirt3 also has no effect on asymmetric division[32]. Interestingly, our data suggest that this lack of effect could reflect compensation by other sirtuins[32]. In keeping with this, protein interaction studies support redundant and complementary roles amongst sirtuins[33]. It is therefore very possible that simultaneous

impairment of multiple NAD-dependent proteins may underpin the phenotype observed when NAD levels are reduced by Nampt-depletion.

Reduced NAD following Nampt-depletion could also derail asymmetric division via altered metabolic redox activity[8,9]. In keeping with this, we find that ATP levels are reduced following

Nampt-depletion and Nampt-inhibition. It is significant in this regard that oocytes are highly dependent upon mitochondrial oxidative phosphorylation for ATP generation[19], and almost half of the oocyte's mitochondria congregate around the spindle perimeter[20]. This spindle-localised mitochondrial pool is proposed to act as a readily accessible ATP fuel-tank to support a highly protracted and energy-demanding acentrosomal spindle assembly and migration process[20,34] in keeping with which, mitochondrial membrane potential in oocytes is maximal in the spindle vicinity[35]. Furthermore, the most energy-demanding events in oocytes appear to occur around the time that anaphase-spindle acceleration occurs as ATP levels and mitochondrial membrane potential peak at the time of PBE[35,36]. Significantly, we find that Nampt co-localises strongly with the mitochondrial pool surrounding the spindle. Moreover, depletion of Nampt reduces peri-spindle levels of both Nampt and mitochondria suggesting that Nampt enrichment around the spindle may reflect mitochondrial Nampt, which has been reported in one paper to be crucial for mitochondrial metabolism[21], albeit this finding has not been replicated in other studies[37]. Reduced mitochondrial abundance could come about, for instance, through compromised function of sirtuins such as Sirt1 and Sirt3, which regulate mitochondrial biogenesis[38]. In line with our findings, knockdown of another salvage pathway enzyme in oocytes, NMNAT2, leads to reduced levels of both NAD and ATP and it was reported that 'oocytes with symmetrical division were frequently observed'[14]. It is also noteworthy that impairing mitochondrial fusion results in markedly enlarged PBs in ~30% of oocytes[39].

Local ATP availability could potentially impact spindle structure and motility through effects on microtubule motors and actin. ATP hydrolysis is required for the function of the kinesin family of microtubule-binding motor proteins[40]. Moreover, disruption of kinesin family members including Kif4[41], HSET (a kinesin-14)[42] and Kif17[43] increase spindle length and in the case of Kif4[44] and Kif17[43], lead to enlarged first PBs. The actin cytoskeleton, which mediates spindle movement in oocytes[1,7], is also heavily dependent upon ATP availability[22–24,45]. However, our work did not identify any alterations in either cortical or cytoplasmic F-actin behaviour following Nampt-depletion, although it remains possible that another actin-dependent process may be involved. One possibility is that altered kinesin functionality resulted in enlarged spindles, which, due to their larger size, were more difficult to displace.

Intriguingly, our data support that Nampt is part of the Mos/MAPK-dependent pathway, which regulates spindle length and division asymmetry in oocytes[7,27]. It is notable in this regard that Nampt function is well-known to overlap with the MAPK pathway in other cell-types[46–49], and MAPK activity has recently been shown to be important for sustaining Nampt levels[30].

Collectively, an appealing model is that Nampt is a downstream mediator of the Mos/MAPK pathway, and regulates an NAD/ATP reservoir critical for spindle assembly and motility, particularly the burst of speed we identify here. We note that although spindle function was disrupted at the relatively modest degree of Nampt-knockdown (~50%) we achieved, meiotic maturation was not, suggesting that oocyte spindles are acutely sensitive to relatively minor changes in NAD/ATP levels that leave maturation intact. This could explain why oocytes assign a dedicated mitochondria-Nampt pool to support spindles. As embryogenesis is critically dependent on cytoplasmic oocyte reserves, these data provide a new perspective on how oocyte metabolism influences developmental competence.

## Methods
**Oocyte isolation and culture**. Animals were housed in filter-top cages in a specific pathogen-free environment and fed a standard diet. All work involving animals complied with all relevant ethical regulations and was approved by the Animal Ethics Committee at the University of Queensland. Mice were allowed ad libitum access to food and water and maintained in a facility with a 12 h light/dark cycle and a 50% relative humidity at 22–24 °C. Ovaries were isolated from 3 to 4-week-old B6CBAF1 female mice 44–46 h following intra-peritoneal injection of 7.5 international units (IU) of pregnant mare's serum gonadotrophin (PMSG; Pacificvet). Dissected ovaries were transferred to the lab in pre-warmed αMEM HEPES-buffered medium (Sigma-Aldrich) containing 50 μM 3-isobutyl-1-methylxanthine (IBMX; Sigma-Aldrich), which prevents oocytes from undergoing GVBD[7,50,51]. Ovaries were punctured in IBMX-treated αMEM HEPES-buffered medium in 35 × 10 mm dishes using a 27 G needle under direct vision on the stage of a stereo-microscope (M165C, Leica Microsystems). Only fully grown cumulus-covered oocytes were isolated, denuded by mouth pipette and used for further experiments. For longer term culture and for all confocal imaging, oocytes were cultured in micro-drops of M16 media (Sigma-Aldrich) under embryo-tested light mineral oil (Sigma-Aldrich) at 37 °C in an atmosphere of 5% $CO_2$ in air.

**Microinjection**. Microinjection was performed as described previously[7]. Microinjection needles were pulled from capillary tubes (0.86 mm inner diameter, 1.5 mm outer diameter; Harvard Apparatus) to a pre-determined calibre using a vertical pipette puller (P30 vertical micropipette puller, Sutter Instruments). For microinjection, GV-stage oocytes in IBMX-treated αMEM HEPES-buffered medium were stabilised using suction applied through a hydraulic syringe to a pre-fabricated holding pipette (inner diameter 15 μm, outer diameter 75 μm, 35° bend; The Pipette Company). The tip of the microinjection pipette was advanced across the zona pellucida and oolemma into the cytoplasm of the oocyte aided by a brief electrical pulse delivered by an intracellular electrometer (IE-25IA, Warner Instruments). A precise volume roughly equal to 5% of the oocyte volume of test solution was delivered to the oocyte using a Pneumatic PicoPump (PV-820, World Precision Instruments). The rate of oocyte death following microinjection was consistently <10%.

**cRNA constructs**. Histone 2B (H2B)-RFP, UtrCH-mCherry (a gift from William Bement[25]; Addgene plasmid #26740) and Sirt2-H187Y-GFP (a gift from Eic Verdin[17]) cRNAs were used. All plasmids were sequenced with T3 primer (Supplementary Table 1) before transcription. The mMESSAGE mMACHINE High Yield Capped RNA Transcription kit (Ambion) was used to produce cRNA constructs by T3-promoter driven in vitro transcription from linearised DNA template[7,51,52]. Following in vitro transcription, cRNA size was verified on agarose gels and concentrations were determined using a spectrophotometer. Constructs were microinjected at the following concentrations: H2B-RFP at 250 ng μl$^{-1}$, UtrCH-mCherry at 700 ng μl$^{-1}$ and Sirt2-H187Y at 1000 ng μl$^{-1}$. Following microinjection at the GV-stage, oocytes were maintained arrested in IBMX-treated αMEM HEPES-buffered medium for at least 2 h to allow time for protein translation before washing free from IBMX to allow maturation to occur.

**Morpholinos**. We used morpholinos for depleting Nampt as before[7,50–53]. GV-stage oocytes were microinjected with a previously validated morpholino sequence designated NamptMO that was designed to target *murine Nampt* (NM_021524.2, ; 5′–CTTCTGCCGCAGCATTCATCTCG–3′) (GeneTools). NamptMO was injected at a final needle concentration of 1.5 mM.

For depleting Mos, GV-stage oocytes were microinjected with a previously validated morpholino sequence designated MosMO that was designed to target *murine Mos* (NM_020021.3; 5′–CACAGGCTTAGAGGCGAAGGCATT–3′) (GeneTools). MosMO was injected at a final needle concentration of 3 mM.

For depleting Sirt2, GV-stage oocytes were microinjected with a previously validated morpholino sequence designated Sirt2MO that was designed to target *murine Sirt2* (NM_022432.4, NM_001122765.1 and NM_001122766.1; 5′–TCGGGACTGTCACCG ACTGCTCTGT–3′) (Gene Tools)[18]. Sirt2MO was injected at a final needle concentration of 2 mM.

For mock-depletion, oocytes were microinjected with a standard control morpholino (designated ControlMO, 5′–CCTCTTACCTCAGTTACAATT TATA–3′)[7,50–53].

Following microinjection, oocytes were maintained in IBMX-treated M16 medium for at least 20 h to allow time for protein knockdown.

**Drug additions, Nampt over-expression and NMN rescue**. For inhibiting Nampt enzymatic activity, FK866, the highly specific non-competitive inhibitor of Nampt[15] (ApexBio/Assay Matrix; 10 mM stock solution in DMSO) was dissolved in medium to a final concentration of 1 μM. Another highly specific Nampt inhibitor, STF-118804[16] (Merck) was dissolved in medium to a final concentration of 500 nM. DMSO was added to medium at the same concentration as was attained when FK866 or STF-118804 was added. GV-stage oocytes maintained in IBMX-treated M16 medium were treated with either DMSO, FK866 or STF-118804 for 24 h before either being lysed for ATP measurements, western blotting or washed into IBMX-free M16 medium containing either DMSO, FK866 or STF-118804 to allow resumption of maturation.

For over-expressing Nampt, recombinant Nampt protein (Visfatin, AdipoGen)[54] wasmicroinjected into oocytes at a concentration of 50 μg ml$^{-1}$.

For NMN rescue, NMN (Sigma-Aldrich) was dissolved in water and was co-injected with NamptMO into oocytes at a concentration of 500 μM.

**Immunoblotting**. Western blotting was performed as described previously[50,51,53]. For sample collection, oocytes were washed in PBS, lysed in LDS sample buffer (NuPAGE; Invitrogen) and snap-frozen and stored at −80 °C until used. For blotting, samples were thawed on ice and boiled for 95 °C for 5 min after adding reducing agent (NuPAGE; Invitrogen). Proteins were separated on 4-12% Bis-Tris gels (NuPAGE; Invitrogen) for 55 min at 200 V in MOPS running buffer (50 mM MOPS, 50 mM Trizma Base, 0.1% SDS and 1 mM EDTA, pH 7.7). Then proteins were transferred to PVDF membranes (Immobilon-P, Millipore) in transfer buffer (0.192 M Glycine, 25 mM Trizma Base and 20% Methanol). Following transfer, membranes were blocked for 1 h at room temperature in 3% BSA in TBS (25 mM Tris, 150 mM NaCl, pH 8.0) containing 0.05% Tween. Membranes were then incubated overnight at 4 °C with primary antibody in blocking solution followed by either an HRP-conjugated goat anti-mouse or goat anti-rabbit antibody (1:1000; Bio-Rad) as the second layer. Primary antibodies used were: rabbit anti-Nampt (1:500; AdipoGen-AG-25A-0028)[54], rabbit anti-Acetylated-α-Tubulin (1:1000; Cell Signaling Technology-5335S), rabbit anti-Sirt2 (1:500; Sigma-Aldrich-S8447), mouse anti-Vinculin (1:2000; Sigma-Aldrich-V9131) and mouse anti-Actin (1:1000; Millipore-MAB1501R). HRP-conjugated secondary antibodies were detected using the Western Lightning ECL-Pro chemiluminescence detection system (Perkin Elmer) and protein bands were imaged using the ImageQuant LAS 500 (GE Healthcare).

To quantify protein levels, we calculated the ratio of the intensity of protein bands to the intensity of the corresponding vinculin loading control and normalised these against values of either uninjected control oocytes (Fig. 1c; Supplementary Fig. 1), ControlMO-injected oocytes (Fig. 7g; Supplementary Fig. 4a) or water-injected oocytes (Supplementary Fig. 4a). Actin and Vinculin served as loading controls to ensure even sample loading and gel transfer. Uncropped blots are shown in source data file.

**Immunofluorescence**. Oocytes were washed in PHEM buffer (pH 7.0) and pre-permeabilised in 0.25% Triton-X in PHEM. Then oocytes were fixed in 3.7% paraformaldehyde solution in PHEM for 20 min. After fixation, oocytes were permeabilised in 0.25% Triton-X in PBS for 15 min. Then, oocytes were blocked overnight in 3% BSA in PBS containing 0.05% Tween-20 at 4 °C. Primary antibody incubation was carried out at 37 °C for 1.5 h. Primary antibodies used were: rabbit anti-Nampt (1:50; AdipoGen-AG-25A-0028) and Rabbit anti-TOM20 (1:200, Santa Cruz-SC-11415). Following three 5-min washes in PBS containing 0.5% BSA and 0.05% Tween-20, oocytes were incubated with the appropriate Alexa Fluor 488-conjugated or 546-conjugated secondary antibodies (1:200; ThermoFisher) for 1 h at 37 °C. Oocytes were then washed three times before staining for DNA by incubating in Hoechst 33342 (10 μg ml⁻¹; Sigma) for 5 min.

To quantify protein levels, fluorescence intensity was determined by measuring the mean background-corrected fluorescence intensity within a region drawn around the spindle. For the plots of Nampt and TOM20 fluorescence, intensity was normalised to the maximum value for each individual oocyte (Fig. 6j, m).

**Measurement of NAD levels**. NAD levels were measured using a commercial kit (Sigma-MAK037) according to the manufacturer protocol. Oocytes from 4 mice per group were used for total NAD extraction. Internal standards were prepared over the range of 0–80 pmol per well (Clear 96-wells plate; Corning). NAD levels were quantified in a colorimetric assay at 450 nm using Spark 10 M Multimode Microplate Reader (Tecan).

**Determination of ATP levels**. The ATP content of individual GV oocyte was determined using the ATP bioluminescent somatic cell assay kit (Sigma). Individual GV oocyte was lysed in 50 μl somatic cell ATP-releasing agent and was stored at −80 °C until further use. A volume of 100 μl diluted ATP assay mix was added to individual wells in an opaque 96-well plate (ThermoFisher). Incubate at room temperature for 5 min to allow endogenous ATP hydrolysis. Internal standards were prepared over the range of 0–1000 fmol per 100 μl. 50 μl of either sample or standards were mixed with 150 μl ATP-releasing agent. 100 μl of the mixture was then transferred to the reaction wells in the opaque 96-well plate and emitted bioluminescence was immediately measured using the Spark 10M Multimode Microplate Reader (Tecan). ATP level in single oocyte was calculated based on the standard curve derived using the internal standards.

**Time-lapse confocal microscopy**. Fluorescence images of spindles and chromosomes as well as brightfield images of oocytes were acquired using a Leica TCS SP8 confocal microscope equipped with a ×20/0.75 NA Apochromat water-immersion objective fitted with an automated pump cap (water-immersion micro dispenser, Leica; automated pump mp-x controller, Bartels Mikrotechnik) as described previously[7].

To visualise microtubules, silicon rhodamine (SiR)-Tubulin dye[7,12] (Cytoskeleton, Inc.) was added to media at a final concentration of 100 nM as validated previously[7].

Oocytes were imaged in 3–4 μl micro-drops of M16 medium in glass-bottom dishes (35 × 10 mm dish, no. 0 coverslip; MatTek), under mineral oil. For the entire duration of imaging, oocytes were enclosed within a purpose-built stage-mounted incubation chamber designed to maintain conditions of 37 °C and 5% CO₂ in air. Temperature fluctuation was further buffered by enclosing the entire microscope, including the stage-mounted chamber, in a custom-designed polycarbonate incubator (Life Imaging Services) that maintained a stable internal temperature of 37 °C. Automated image capture was driven by the Leica LAS X software. At the commencement of imaging, the positions of the spindles in the z-axis were located. The complete stack was then derived by setting a thickness of 50 μm. Z-stacks were acquired with step intervals of 3.5 μm at 5 min intervals at a speed of 600 Hz. Using the Leica mark-and-find tool, we typically imaged multiple groups of oocytes in separate droplets in each experiment. Leica HyD detectors enabled the highly sensitive detection of very low levels of fluorophore emission, ultimately minimising the required laser excitation levels. The 561 and 633 nm laser lines were typically used at 0.5% and 3% power, respectively.

Post-acquisition image processing was performed using Leica LAS X software and images were assembled into panels using Adobe Photoshop. Fluorescence images and movies were produced by merging a maximum projection of the fluorescent channels with or without a single plane of brightfield channel.

**Quantification of spindle dimensions and PB size**. For analysis of spindle migration, only oocytes having bipolar spindles that were oriented in the horizontal plane and that remained in the same orientation throughout anaphase, protrusion and cytokinesis were included in analyses.

Spindle length was measured manually as the pole-to-pole distance along the long axis of the spindle at indicated stages. As the average length of anaphase spindles in water-injected oocytes at 10-min post-anaphase-onset was 33.42 ± 0.71 μm (mean ± SEM), and >83% of the water-injected oocytes had anaphase-spindle lengths ≥32 μm, short anaphase spindles were defined as spindles that were <32 μm at 10 min post-anaphase-onset (Supplementary Fig. 3c).

Spindle width was measured manually, as the maximal distance between the extreme limits of the spindle along a line positioned roughly midway between the two spindle poles and perpendicular to the long axis of the spindle.

During spindle migration, the leading spindle pole was considered the pole that migrated into the oocyte membrane closest to the spindle (termed anterior membrane) following anaphase-onset. The other (non-leading) spindle pole was considered the lagging pole and the membrane closest to the lagging pole was considered the posterior membrane (Figs. 2e and 4a, b). To quantify pre-anaphase-spindle migration, the distance between the leading pole and the anterior oocyte membrane was determined hourly from 4 h prior to anaphase to anaphase-onset (Fig. 2e, f). To eliminate the possibility that different spindle sizes might introduce artefacts in apparent spindle migration, the distance between the middle of the spindle and the anterior oocyte membrane along the long axis of the spindle was also measured (Fig. 2e, g). To quantify post-anaphase-onset spindle migration, the distances between the posterior pole and the posterior oocyte membrane (pole-to-membrane distance) as well as between the midzone and the posterior oocyte membrane (midzone-to-membrane distance) were determined at successive time points throughout anaphase (Fig. 4a–d).

Maximum PB height was determined as the distance from the base of the PB (at the level of the surrounding oocyte membrane) to the most extreme aspect of the out-pocketing determined from the membrane along the line of the long axis of the spindle when the PB height was maximal during anaphase (Fig. 5e, f, h).

PB width was determined as the maximal distance across the extreme limits of the PB along a line positioned roughly midway between the base of the PB and the most extreme aspect of the out-pocketing. PB width was measured after PB extrusion was completed. Data are presented as the ratio of PB width to oocyte diameter at the GV-stage (PB:Oocyte ratio) to control for any small variation in oocyte sizes. As mean PB:Oocyte ratios in uninjected and controlMO-injected oocytes were 42.83 ± 1.04% and 40.63 ± 0.75%, respectively, and >85% of oocytes had ratios <50% in control group, a PB:Oocyte ratio >50% was used to define large PBs (Figs. 1g, 6c, d, f and 7d).

Maximum protrusion or bulge width was determined as the maximal distance across the extreme limits of the base of either the protrusion or the bulge. The width was measured when protrusion or bulge occurred (Fig. 5c).

PB2 height was determined as the distance from the base of the PB2 (at the level of the surrounding oocyte membrane) to the most extreme aspect of the out-pocketing determined along the long axis of the spindle when the PB2 height was maximal. Data are presented as the ratio of PB2 height to oocyte diameter (PB2:Oocyte ratio) to control for any small variation in oocyte sizes. As mean PB2 height:Oocyte ratio in controlMO-injected oocytes was 19.57 ± 0.89% and >90% of oocytes had ratios <23.5% in control group, a PB:Oocyte ratio >23.5% was used to define large PB2s (Supplementary Fig. 2b).

**4D analysis of spindle migration**. Time-lapse data were imported into Imaris software (Bitplane, Switzerland) for detailed analysis of spindle migration. The intensity of SiR-Tubulin staining was used to detect spindle boundaries and Surface Tool was used for 3D reconstruction of the spindle with background subtraction at 10 μm and surface detail of 1 μm. The tracking algorithm, Autoregressive Motion, was used to track spindle movement with a maximum gap distance of 15 μm and

gap size of 3. Spindle speed was determined for each time point as the distance travelled by the centre of the surface over time.

**Quantification of cytoplasmic and cortical F-actin**. UtrCH-mCherry fluorescence intensity was used for quantifying F-actin levels in live oocytes[7,25,26]. Cytoplasmic UtrCH-mCherry intensity was determined by measuring the mean background-corrected fluorescence intensity within a cortex-free region drawn immediately adjacent to the lagging spindle pole. For the plots of UtrCH-mCherry fluorescence, intensity was normalised to the maximum value for each individual oocyte (Supplementary Fig. 5a).

Cortical thickness was measured at the anterior cortical surface closest to the leading pole of the spindle at 5 min prior to anaphase-onset. For comparison, cortical thickness was measured at three other sites away from the anterior cortex: at the cortex opposite to the anterior cortex (I) and at two additional sites between I and the anterior cortex that were roughly opposite from one another (II and III) from which, the mean cortical thickness of non-polarised cortex was determined as done previously[7]. Values shown are anterior cortical F-actin thickness relative to the mean of the thicknesses at positions I, II and III for each oocyte. (Supplementary Fig. 5b, c).

**Statistical analysis**. GraphPad Prism (GraphPad, USA) was used to calculate mean and standard error of the mean (SEM). Statistical comparisons were made using two-tailed Student's $t$-test (for comparing two experimental groups, with Welch's correction if the $P$-value in $F$-test was significant), one-way ANOVA with Tukey's multiple comparisons (for comparing three or more experimental groups), two-way ANOVA with Sidak's multiple comparisons (for comparing the experimental groups that were affected by two factors), or two-way repeated measures ANOVA with Sidak's multiple comparisons (for comparing the experimental groups that were measured at the same time points) as appropriate to the data set. Graphs were prepared in GraphPad Prism. Data in graphs were presented as the mean ± SEM. Oocyte numbers were shown in parenthesis. $P$-values were shown in graphs, ns denoted $P > 0.05$. All experiments were repeated three times.

**Reporting summary**. Further information on research design is available in the Nature Research Reporting Summary linked to this article.

## Data availability

The data that support the findings of this study are available from the corresponding author upon reasonable request. The source data underlying Figs. 1–7 and Supplementary Figs. 1–5 are provided as a Source Data file. Source data are provided with this paper.

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

## Acknowledgements

This work was funded by the Professor Christopher Chen Endowment Fund, Startup funding from the Faculty of Medicine, University of Queensland and National Health and Medical Research Council Project Grants (APP1078134, APP1103689 and APP1122484) to H.A.H. We are grateful to Stephen Thompson (Leica Microsystems) for technical support with confocal imaging.

## Author contributions

H.A.H. conceived the project and wrote the paper. Z.W. undertook experiments, analysed most of the data and prepared figures. J.G. contributed to the experiments, analysed the data using Imaris software and measured the cytoplasmic F-actin intensity. W.-G.N.L. undertook experiments involving Sirt2 knockdown and the Sirt2-mutant construct, Sirt2-H187Y.

## Competing interests

H.A.H. is a co-founder of a research company, JumpStart Fertility, whose intention is to develop agents for improving oocyte quality. All remaining authors declare no competing interests.

## Additional information

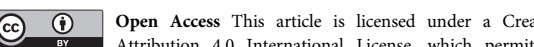

