## [Peer Review File · Nature Communications]

Reviewers' comments:

Reviewer #1 (Remarks to the Author):

In this report, Dr. Wei and collaborators have investigated the effect of nicotinamide phosphoribosyl-transferase (Nampt) on the migration of the spindle to the cortex during oocyte meiosis. The authors show that Nampt KD disrupts the highly asymmetrical oocyte division, yielding a significantly larger polar body (PB). This phenotype is not associated with altered movements of the spindle toward the oocyte cortex. However, bulging of the membrane is prevented in oocytes depleted of Nampt. Pharmacological inhibition of Nampt does not recapitulate the phenotype suggesting that Nampt catalytic activity may not be required nor is the altered tubulin alpha acetylation affecting spindle movements. Given that the phenotype of enlarged polar body is reminiscent of the Mos KO phenotype, the authors explore a possible connection between the two pathways and show that Mos KO leads to a decrease in Nampt and Nampt overexpression partially rescues the phenotype associated with Mos depletion.

The study provides interesting information on spindle assembly and the mechanisms underlying the asymmetrical division of a mouse oocyte. While these observations are relevant, they are quite preliminary and descriptive. The authors do not provide mechanistic data to establish a causative link between the loss of Nampt function and the phenotypes that they describe in considerable detail. Although one could assume that disruption of the NAD salvage pathway is at the root of the phenotype, the experiment suggesting that NAMPT catalytic activity is not required rejects this view. This pharmacological manipulation may have not been correctly controlled. The observation that Mos depletion affects Nampt expression is interesting but again not developed in terms of molecular mechanisms.

1. One measure that the authors use is the ratio of the PB diameter versus the oocyte diameter. Since it is likely that oocyte diameter decreases as the division becomes less asymmetrical, it is not an absolute measure of the PB size. This reviewer is wondering whether a more meaningful (and perhaps more stable) number would be the PB volume not corrected for the oocyte volume (ImageJ-NIH?). Statistical data on the rescue comparison control MO Namt MO+ Nampt protein is not reported. Also not included is a Western blot comparing the amount of recombinant Nampt protein injected with the endogenous protein. This information is necessary for the reader to verify whether Nampt levels are restored to normal or if the recombinant Nampt is in large excess over the endogenous one.
2. The authors use FK866 to inhibit the Nampt enzymatic activity. To determine whether the pharmacological treatment is effective they use acetylated tubulin as a readout. The problem is that the changes in acetylated tubulin detected are very small (10-20% of the overall signal). Given this poor dynamic range, it is not clear whether this readout is sufficiently sensitive to monitor the effect of the pharmacological inhibition. As a consequence, it is unclear whether the conclusion that the Namt enzymatic activity is not required is correct. Although very effective at subnanomolar concentrations in tumor inhibition, assays of inhibition of NAMPT activity in human HepG2 cells using ¹⁴C-nicotinamide mononucleotide indicate an IC₅₀ of 2.2 nM. Therefore, it is not clear whether the 25 nM used is sufficient to completely inhibit the activity of the enzyme in intact cells. Also, the authors should consider that the oocytes have a large extracellular matrix (zona pellucida) that limits diffusion. The authors should have tested the inhibition of Nampt in intact oocytes using labeled substrate and enzymatic assays (PMID24164086) before they can conclude that NAMPT activity is completely blocked under their conditions. Does Nampt KD affect NAD levels in the oocyte?
3. Given that the phenotype of enlarged polar body is reminiscent of the Mos KO phenotype, the authors explore a possible connection between the two pathways and show that Mos depletion causes a decrease in Nampt. Although an interesting observation, conclusions are difficult to draw. Mos depletion has major effects on numerous signaling cascades in the oocytes, rendering it difficult to accept the conclusion that the Mos is epistatic to Nampt. The authors do not state whether activated oocytes in the Mos-depleted group are included in the measurements. The effect

of Nampt expression on a Mos-null background are rather small, and reversal is incomplete. Twice as many oocytes were used in the Mos KD+ Nampt group to reach statistical significance.

4. The authors conclude that Nampt has a function in spindle migration and bulging of the membrane but do not provide a molecular explanation on how Nampt is involved. Do the authors envision that the Nampt protein is interacting with spindle or cytoskeletal components? Is it acting as a scaffold

Reviewer #2 (Remarks to the Author):

The major claims of the manuscript by Wei et al. is that (a) the metabolic enzyme Nampt participates in the asymmetric division of mouse oocytes and (b) the degree of asymmetry depends on the speed of spindle movement following the onset of anaphase. The manuscript is very well structured and detailed. In addition, it provides significant novelty by introducing a metabolic enzyme in the regulation of asymmetric division and by identifying the speed of anaphase spindle movement as a major contributor to asymmetry. Nevertheless, these hypotheses might need more explanation supported by further experiments in order to be considered solid facts.

1. The authors claim that anaphase spindle movement is the main cause of the asymmetric division in MI oocytes. However, the position of the midzone determines the site of the cleavage furrow and therefore the symmetry or asymmetry of cell division. In MI oocytes, the spindle is placed perpendicularly to the plasma membrane. A longer spindle would possess a more central midzone, while a smaller spindle will possess a midzone closer to the cortex. Therefore, a longer spindle will cause a deeper furrow compared to the superficial furrow of a short spindle. Is there any reason to believe that this is not the main reason of asymmetry loss in the Nampt-depleted or Mos-depleted oocytes? Even if the spindle speed was the same in controls and Nampt-depleted oocytes, wouldn't the Nampt-depleted oocyte midzone still be less cortically located compared to controls causing a deeper furrow and therefore a larger polar body?

2. The formation of the actin cap and PAR protein spindle localization, which are the main determinants of asymmetric division, occur prior to anaphase. Do actin and PAR proteins show the same localization prior to anaphase in control and Nampt-depleted oocytes? If they do not, one could argue that the decision for a large polar body has been taken before anaphase, in the Nampt-depleted oocytes.

3. The authors claim that the Nampt-depleted oocytes have a bulge instead of a protrusion. However, one might argue that a protrusion is not absent in Nampt-depleted oocytes, but instead that the protrusion is wider because the furrow is deeper due to the more centrally placed midzone of the long spindle. How do we define the protrusion and when is a bulge or projection not considered a protrusion?

4. It is suggested that the presence of Nampt and not its activity is responsible for asymmetric division. Are there known functions of Nampt that are not related to its enzymatic activity? How could just the presence of Nampt explain asymmetry? Are there other pharmacological inhibitors of Nampt, besides FK866, that could be used to show that the enzyme's activity is not related to asymmetry?

5. In order to verify further the role of Nampt, is it possible for the authors to perform immunofluorescence experiments for Nampt in MI oocytes in order to show its localization in the vicinity of cell division?

6. Nampt enzymatic activity does not affect asymmetry and Mos depletion causes loss of the Nampt protein. Could the authors explain how the Mos/MAPK pathway of kinases may affect

Nampt accumulation or stability? Is there bibliographical or experimental evidence of how the Mos/MAPK pathway affects Nampt protein levels?

7. Since Mos and Nampt participate in the same pathway, they would affect polar body extrusion in the same way. Is there evidence that Mos depletion causes a delay in anaphase spindle movement and a single furrowing site, as seen in Nampt-depleted oocytes?

8. Is the Nampt depletion phenotype seen in Meiosis II?

Reviewer #3 (Remarks to the Author):

Wei, Greaney, and Homer

"Nampt-mediated spindle sizing secures a post-anaphase surge in spindle speed required for extreme asymmetry"

In this study, the authors showed that Nampt depletion in mouse oocytes caused the extension of pre-anaphase and anaphase bipolar spindles and the slowdown of anaphase spindle migration, resulting in the formation of larger polar bodies (PBs). The phenomenon that the authors discovered is very interesting and could be important to understand the physiological importance of Nampt-mediated NAD⁺ biosynthesis in mammals. However, mechanistic details remain unclear, and even the involvement of NAD⁺ in this observed phenomenon was not fully investigated in this study. Therefore, although this study may be a significant interest to general readers, more mechanistic details should be further elaborated before being considered for publication.

Major points:

1. When knocking down Nampt, are NAD⁺ levels in mouse oocytes decreased? Direct measurement of NAD⁺ in mouse oocytes would be technically challenging. Nonetheless, there are NAD⁺ sensors available (e.g. the one that Dr. Lulu Cambrone's lab is working on: <https://www.lulucambronne.org/publications>). Thus, it is very important to use these new NAD⁺ sensors and demonstrate changes in NAD levels in mouse oocytes when depleting Nampt.

2. Related to the point above, if the regulation of pre-anaphase and anaphase bipolar spindles is NAD⁺-dependent, the authors should conduct a rescue experiment using the product of the Nampt enzymatic reaction, namely, nicotinamide mononucleotide (NMN). It is important to examine whether NMN can rescue all these defects described in this study. Whether NMN could work or not may be dependent on the type of media (please check the concentration of nicotinamide in the media). If nicotinamide concentration is unphysiologically high (for example, more than 10 microM), NMN may not work. In that case, the authors may want to use nicotinic acid for the rescue experiment.

3. The authors' FK866 experiment is inconclusive because it is likely that the concentration of FK866 that they used is too low (25 nM). Primary mouse cells tend to require high concentrations of FK866. For example, 500 nM FK866 is necessary to effectively inhibit Nampt in primary hepatocytes. Therefore, the authors should check higher concentrations of FK866 as far as those concentrations are not toxic to mouse oocytes. If the toxicity becomes an issue, the Nampt knockdown is better. In any case, the authors should check NAD⁺ levels in all these experiments.

4. It is also critical to elaborate what the effector would be in response to NAD⁺ changes. A usual suspect is probably one of the mammalian sirtuin family members, most likely Sirt2 (Please type in "Sirt2 AND spindle formation" to the PubMed. Many related publications can be pulled by these search terms). The authors should examine whether knocking down Sirt2 (or other sirtuins) could possibly cause similar phenotypes in mouse oocytes. If the rescue experiment works, the authors

could also examine whether Sirt2 (or other sirtuins) may be required for the rescue. Then, it is important to examine whether tubulin acetylation is changed in mouse oocytes when depleting Sirt2 (or other sirtuins).

Minor points:

1. Where is the Nampt protein localized in mouse oocytes? Have the authors conducted the immunostaining of Nampt in mouse oocytes?
2. For the data to which the authors applied two-way ANOVA, the two-way repeated-measures ANOVA would be appropriate.
3. The Discussion section has many speculations. The authors should address the major points described above, and based on the results, the authors should revise their discussion.

We wish to submit the revised version of the above-named paper for consideration for publication in *Nature Communications*. We are very grateful for the constructive criticism provided by the reviewers. Below, we provide a detailed point-by-point response to the reviewer's comments. Please note that comments from the reviewers are in italics.

Reviewer #1 (Remarks to the Author):

In this report, Dr. Wei and collaborators have investigated the effect of nicotinamide phosphoribosyl-transferase (Nampt) on the migration of the spindle to the cortex during oocyte meiosis. The authors show that Nampt KD disrupts the highly asymmetrical oocyte division, yielding a significantly larger polar body (PB). This phenotype is not associated with altered movements of the spindle toward the oocyte cortex. However, bulging of the membrane is prevented in oocytes depleted of Nampt. Pharmacological inhibition of Nampt does not recapitulate the phenotype suggesting that Nampt catalytic activity may not be required nor is the altered tubulin alpha acetylation affecting spindle movements. Given that the phenotype of enlarged polar body is reminiscent of the Mos KO phenotype, the authors explore a possible connection between the two pathways and show that Mos KO leads to a decrease in Nampt and Nampt overexpression partially rescues the phenotype associated with Mos depletion.

The study provides interesting information on spindle assembly and the mechanisms underlying the asymmetrical division of a mouse oocyte. While these observations are relevant, they are quite preliminary and descriptive. The authors do not provide mechanistic data to establish a causative link between the loss of Nampt function and the phenotypes that they describe in considerable detail. Although one could assume that disruption of the NAD salvage pathway is at the root of the phenotype, the experiment suggesting that NAMPT catalytic activity is not required rejects this view. This pharmacological manipulation may have not been correctly controlled. The observation that Mos depletion affects Nampt expression is interesting but again not developed in terms of molecular mechanisms.

1. One measure that the authors use is the ratio of the PB diameter versus the oocyte diameter. Since it is likely that oocyte diameter decreases as the division becomes less asymmetrical, it is not an absolute measure of the PB size. This reviewer is wondering whether a more meaningful (and perhaps more stable) number would be the PB volume not

corrected for the oocyte volume (ImageJ-NIH?).

We agree entirely that if we had performed the ratio between the PB and the secondary oocyte (that is, the oocyte remnant after PBE), then the value obtained for PB size would be artificially inflated due to the relatively low size of the secondary oocyte. However, the ratio was performed relative to the oocyte diameter at the GV-stage and therefore PRIOR to asymmetric division as done previously (Wei et al. 2018 Nat Commun). We apologise for not making this clearer in the original submission and have now included these important details in the revised Methods.

Statistical data on the rescue comparison control MO Namt MO+ Nampt protein is not reported.

We now include the relevant statistical comparisons in the graphs as requested (Fig. 1g, h).

Also not included is a Western blot comparing the amount of recombinant Nampt protein injected with the endogenous protein. This information is necessary for the reader to verify whether Nampt levels are restored to normal or if the recombinant Nampt is in large excess over the endogenous one.

We agree with the reviewer that this is an important point. We have therefore undertaken these Westerns as requested and find that Nampt is not being expressed in large excess over endogenous levels (Supplementary Fig. 1).

2. The authors use FK866 to inhibit the Nampt enzymatic activity. To determine whether the pharmacological treatment is effective they use acetylated tubulin as a readout. The problem is that the changes in acetylated tubulin detected are very small (10-20% of the overall signal). Given this poor dynamic range, it is not clear whether this readout is sufficiently sensitive to monitor the effect of the pharmacological inhibition. As a consequence, it is unclear whether the conclusion that the Namt enzymatic activity is not required is correct. Although very effective at subnanomolar concentrations in tumor inhibition, assays of inhibition of NAMPT activity in human HepG2 cells using 14C-nicotinamide mononucleotide indicate an IC50 of 2.2 nM. Therefore, it is not clear whether the 25 nM used is sufficient to completely inhibit the activity of the enzyme in intact cells. Also, the authors should consider that the oocytes have a large extracellular matrix (zona pellucida) that limits diffusion. The authors should have tested the inhibition of Nampt in intact oocytes using labeled substrate and enzymatic assays (PMID24164086) before they can conclude that NAMPT activity is completely blocked under their conditions. Does Nampt KD affect NAD levels in the oocyte?

The reviewer very rightly challenges whether we achieved substantial inhibition of Nampt at the dose of FK866 (25 nM) we used in our initial experiments. Since as reviewer 3 points out, 500 nM is required in primary hepatocytes and as reviewer 1 points out, the unique properties of oocytes likely require even higher concentrations, we used 1 μ M FK866. Importantly, this level did not impair meiotic maturation but did significantly reduce NAD levels and markedly disrupted asymmetric division (Fig. 6b, c). We further showed that use of another highly specific Nampt inhibitor, STF-118804 (Matheny et al. 2013 Chem Biol), also reduced NAD levels and led to compromised asymmetry (Fig. 6b, d). We also found using NAD assays that Nampt-depletion also reduces NAD levels (Fig. 6a) and importantly, that NMN (the product of the Nampt enzymatic reaction in the salvage pathway for generating NAD) rescues defects in spindle lengths and PB size (Fig. 6e, f).

3. Given that the phenotype of enlarged polar body is reminiscent of the Mos KO phenotype, the authors explore a possible connection between the two pathways and show that Mos depletion causes a decrease in Nampt. Although an interesting observation, conclusions are difficult to draw. Mos depletion has major effects on numerous signaling cascades in the oocytes, rendering it difficult to accept the conclusion that the Mos is epistatic to Nampt. The authors do not state whether activated oocytes in the Mos-depleted group are included in the measurements.

The reviewer asks whether activated oocytes were included in the measurements. However, spontaneous activation is a phenomenon that occurs sometime after MII-arrest is established and therefore AFTER the event we are interested in has occurred – this event of interest being first polar body extrusion at the end of MI. We fully expect that a proportion of Mos-depleted oocytes would have spontaneously activated after entering MII as shown before (Coonrod et al. 2001 Genesis; Nabti et al. 2014 J Cell Biol; Wei et al. 2018 Nat Commun), but we did not extend our analyses that long to identify such a phenomenon as it was not relevant to the main focus of this paper.

The effect of Nampt expression on a Mos-null background are rather small, and reversal is incomplete. Twice as many oocytes were used in the Mos KD+ Nampt group to reach statistical significance.

We have increased the numbers of oocytes in the ControlMO and MosMO groups so the three groups now number 36+41+39 for spindle length analyses and 43+53+57 for PB size analyses and find that the level of significance is not reduced, and indeed, in some cases, is even greater (Fig. 7b-d).

4. The authors conclude that Nampt has a function in spindle migration and bulging of the membrane but do not provide a molecular explanation on how Nampt is involved.

In the revised version we have undertaken an extensive range of additional experiments, which support possible mechanisms by which Nampt may impact spindle length and motility. These new experiments show that both NAD and ATP levels are reduced following Nampt-depletion (Fig. 6a, g). We therefore propose that at least one mechanism by which Nampt-depletion can impact spindles is via a metabolic NAD/ATP defect, which may have a marked effect on spindles given our new finding of strong Nampt co-localisation with mitochondria around the spindle periphery and the disruption of both following Nampt-depletion (Fig. 6h-l).

Do the authors envision that the Nampt protein is interacting with spindle or cytoskeletal components? Is it acting as a scaffold

As mentioned above, we have undertaken new immunostaining experiments in the revised manuscript and find that Nampt and mitochondria co-localise strongly around the spindle periphery and significantly, that both are compromised following Nampt-depletion (Fig. 6h-l). Aside from the metabolic defect secondary to reduced NAD/ATP we identify (Fig. 6a, g), we therefore do not discount that Nampt/mitochondria may play a scaffolding role that supports spindles as suggested by the reviewer.

Reviewer #2 (Remarks to the Author):

The major claims of the manuscript by Wei et al. is that (a) the metabolic enzyme Nampt participates in the asymmetric division of mouse oocytes and (b) the degree of asymmetry depends on the speed of spindle movement following the onset of anaphase. The manuscript is very well structured and detailed. In addition, it provides significant novelty by introducing a metabolic enzyme in the regulation of asymmetric division and by identifying the speed of anaphase spindle movement as a major contributor to asymmetry. Nevertheless, these hypotheses might need more explanation supported by further experiments in order to be considered solid facts.

1. The authors claim that anaphase spindle movement is the main cause of the asymmetric division in MI oocytes. However, the position of the midzone determines the site of the cleavage furrow and therefore the symmetry or asymmetry of cell division. In MI oocytes, the spindle is placed perpendicularly to the plasma membrane. A longer spindle would possess a more central midzone, while a smaller spindle will possess a midzone closer to the cortex. Therefore, a longer spindle will cause a deeper furrow compared to the superficial furrow of a short spindle. Is there any reason to believe that this is not the main reason of asymmetry loss in the Nampt-depleted or Mos-depleted oocytes?

This very important issue raised by the reviewer is exactly the point we are making regarding protrusion-mediated and non-protrusion pathways. In the former case, which occurs in >90% of controls (Fig. 5a, b), the protrusion delivers half of the spindle beyond the oocyte boundary (Fig. 5a). Furrowing then occurs at the base of the protrusion, the position of which, coincides with the oocyte surface (see Fig. 5a). In other words, during the normal protrusion-mediated pathway, furrowing does NOT occur deep within the oocyte but at the base of the protrusion (i.e. the oocyte surface). Since furrowing in this instance roughly coincides with the position of the geometric middle of the midzone (Fig. 5d, +ControlMO), the height of the protrusion is directly proportional to spindle length (Fig. 5e, f). Importantly, in this pathway, furrowing always occurs at the protrusion base (i.e. the oocyte surface) and greater spindle length causes greater protrusion height rather than causing the furrowing position to be pushed deeper into the oocyte.

In contrast, in the non-protrusion pathway (Nampt-depleted oocytes), protrusion formation is compromised and furrowing now occurs deep within the oocyte. The much larger calibre of oocyte cleaved off in this situation leads to larger PBs (Fig. 5g).

Even if the spindle speed was the same in controls and Nampt-depleted oocytes, wouldn't the Nampt-depleted oocyte midzone still be less cortically located compared to controls causing a deeper furrow and therefore a larger polar body?

We predict that if post-anaphase-onset surge in spindle speed were to be sustained in Nampt-depleted oocytes, then a protrusion-type pathway would occur because furrowing would be delayed due to rapid motion of the midzone thereby furnishing time for a protrusion to form (Wei et al. 2018 Nat Commun). In this case, a protrusion would form and as explained above, furrowing would occur at the protrusion base (i.e. at the oocyte surface). Nampt-depleted oocytes would nevertheless create a larger PB than control oocytes because the PB height would be increased proportionate to the spindle length (Fig. 5e, f) whilst the protrusion width would remain stable (see Fig. 5c, +ControlMO). Significantly, however, PBs formed through this pathway would be expected to be smaller than those formed in a non-protrusion pathway

because in the latter, furrowing is occurring through the large-diameter oocyte rather than at the base of a protrusion tightly moulded around the spindle. We were not able to formally test this since there was no occasion in which, very long spindles in Nampt-depleted oocytes exhibited a post-anaphase-onset surge in speed.

2. The formation of the actin cap and PAR protein spindle localization, which are the main determinants of asymmetric division, occur prior to anaphase. Do actin and PAR proteins show the same localization prior to anaphase in control and Nampt-depleted oocytes? If they do not, one could argue that the decision for a large polar body has been taken before anaphase, in the Nampt-depleted oocytes.

The reviewer asks an important question regarding localised pre-anaphase cortical remodelling, which is important for pre-anaphase spindle migration. In the revised paper, we undertook live imaging of UtrCH-mCherry to monitor cortical actin and found that cortical thickening immediately pre-anaphase was the same in controls as in Nampt-depleted oocytes (Supplementary Fig. 5b, c) entirely consistent with intact pre-anaphase spindle migration (Fig. 2).

We used a commercial anti-PAR3 antibody used previously (Duncan et al. 2005 Dev Biol) but were not able to obtain convincing cortical or spindle localisation in either controls or Nampt-depleted oocytes prior to anaphase.

3. The authors claim that the Nampt-depleted oocytes have a bulge instead of a protrusion. However, one might argue that a protrusion is not absent in Nampt-depleted oocytes, but instead that the protrusion is wider because the furrow is deeper due to the more centrally placed midzone of the long spindle. How do we define the protrusion and when is a bulge or projection not considered a protrusion?

We agree with the reviewer that protrusions should be better defined. We have detailed above how protrusion formation comes about and that furrowing occurs at the base of the protrusion (i.e. at the oocyte surface). The width of protrusions is very stable irrespective of spindle length because protrusions are moulded around spindles and spindle widths are relatively constant (Fig. 5c, +ControlMO). We have undertaken detailed measurements in control oocytes that enabled us to define protrusions in the revised manuscript as membrane out-pocketings that are moulded around spindles and consequently, whose maximal dimensions are <2-fold the anaphase- spindle width and roughly half the anaphase- spindle length (Fig. 5c, d).

For bulges, in contrast, rather than being a true membrane out-pocketing induced by the spindle, membrane ingression into the oocyte is partly responsible for giving the appearance of bulging beyond the narrowed region (Fig. 5g). Consequently, bulges are not moulded around spindles with the result that the widths of bulges are markedly larger than protrusions (Fig. 5c) and their heights show little correlation with spindle length (Fig. 5h). Notably, the large widths of bulges in Nampt-depleted oocytes despite spindle widths being no different from controls (Figs. 3e and 5c), reaffirming that bulges are distinct from protrusions moulded around spindles.

4. It is suggested that the presence of Nampt and not its activity is responsible for asymmetric division. Are there known functions of Nampt that are not related to its enzymatic activity? How could just the presence of Nampt explain asymmetry?

Based on comments from Reviewers 1 and 3, we have revisited experiments designed to inhibit Nampt and find that NAD levels are in fact reduced in Nampt-depleted oocytes when adequate doses of small molecule Nampt inhibitors are employed. These reviewers pointed out that the dose of FK866 (25 nM) we used in our initial experiments was likely too low to inhibit Nampt in oocytes. Since as reviewer 3 points out, 500 nM is required in primary hepatocytes and as reviewer 1 points out, the unique properties of oocytes likely require even higher concentrations, we used 1 μ M FK866. Importantly, this level did not impair meiotic maturation but did significantly reduce NAD levels and markedly disrupted asymmetric division (Fig. 6b, c).

We have undertaken new immunostaining experiments and find that Nampt and mitochondria co-localise strongly around the spindle periphery and that both are compromised following Nampt-depletion (Fig. 6h-l). We also find that ATP levels are reduced following Nampt-depletion (Fig. 6g). Aside from the metabolic defect related to reduced NAD and ATP levels, we therefore do not discount that Nampt/mitochondria may play a scaffolding role to support spindles as suggested by Reviewer 1.

Are there other pharmacological inhibitors of Nampt, besides FK866, that could be used to show that the enzyme's activity is not related to asymmetry?

As explained above, our new experiments with higher doses of Nampt inhibitors in conjunction with NAD assays show that reduced NAD levels are in fact an important cause for asymmetry when Nampt is depleted (Fig. 6a-g). To support that reduced NAD levels are important, we showed that another highly specific Nampt inhibitor, STF-11804 (Matheny et al. 2013 Chem Biol), also reduced NAD levels and led to compromised asymmetry (Fig. 6b, d). Furthermore, we also found using NAD assays that Nampt-depletion reduces NAD levels (Fig. 6a) and importantly, that NMN (the product of the Nampt enzymatic reaction in the salvage pathway for generating NAD) rescues defects in spindle lengths and PB size (Fig. 6e, f).

5. In order to verify further the role of Nampt, is it possible for the authors to perform immunofluorescence experiments for Nampt in MI oocytes in order to show its localization in the vicinity of cell division?

We agree that this is an important experiment. As mentioned above, we have now conducted new immunolocalization experiments and find that Nampt exhibits a striking localisation around the spindle perimeter that coincides with mitochondrial enrichment (Fig. 6h-j). Moreover, when Nampt is depleted, both Nampt and mitochondrial localisation around the spindle are compromised (Fig. 6h-l).

6. Nampt enzymatic activity does not affect asymmetry and Mos depletion causes loss of the Nampt protein. Could the authors explain how the Mos/MAPK pathway of kinases may affect Nampt accumulation or stability? Is there bibliographical or experimental evidence of how the Mos/MAPK pathway affects Nampt protein levels?

As explained above, our new experiments with higher doses of Nampt inhibitors in conjunction with NAD assays show that reduced NAD levels are in fact an important cause for asymmetry when Nampt is depleted. We would also like to point out that inhibiting MAPK in melanoma cells leads to reduced Nampt levels thereby providing independent evidence that MAPK regulates Nampt (Ohanna et al. 2018 Genes Dev).

7. *Since Mos and Nampt participate in the same pathway, they would affect polar body extrusion in the same way. Is there evidence that Mos depletion causes a delay in anaphase spindle movement and a single furrowing site, as seen in Nampt-depleted oocytes?*

We provide new data in the revised manuscript showing that the surge in spindle speed observed after anaphase-onset in controls is severely compromised in Mos-depleted oocytes and that there is an increased proportion of Mos-depleted oocytes with a unilateral furrowing site (Fig. 7e, f).

8. *Is the Nampt depletion phenotype seen in Meiosis II?*

Our objective was to analyse asymmetric division in MI. PBE in MII involves different processes since the spindle is already cortically located after MI, lies parallel to the cortex and must undergo rotation (e.g. Wang et al. 2011 Cell Div), added to which calcium signalling is required to break metaphase II arrest. It was not feasible for us to repeat all our experiments in the unique context of MII but we did find that when MII-arrested Nampt-depleted oocytes were activated, the size of the second PB was increased, which we report (Supplementary Fig. 2) but do not delve into further.

Reviewer #3 (Remarks to the Author):

Wei, Greaney, and Homer

“Nampt-mediated spindle sizing secures a post-anaphase surge in spindle speed required for extreme asymmetry”

In this study, the authors showed that Nampt depletion in mouse oocytes caused the extension of pre-anaphase and anaphase bipolar spindles and the slowdown of anaphase spindle migration, resulting in the formation of larger polar bodies (PBs). The phenomenon that the authors discovered is very interesting and could be important to understand the physiological importance of Nampt-mediated NAD⁺ biosynthesis in mammals. However, mechanistic details remain unclear, and even the involvement of NAD⁺ in this observed phenomenon was not fully investigated in this study. Therefore, although this study may be a significant interest to general readers, more mechanistic details should be further elaborated before being considered for publication.

Major points:

1. *When knocking down Nampt, are NAD⁺ levels in mouse oocytes decreased? Direct measurement of NAD⁺ in mouse oocytes would be technically challenging. Nonetheless, there are NAD⁺ sensors available (e.g. the one that Dr. Lulu Cambrone's lab is working on: <https://www.lulucambronne.org/publications>). Thus, it is very important to use these new NAD⁺ sensors and demonstrate changes in NAD levels in mouse oocytes when depleting Nampt.*

This is a very important point raised by the reviewer. As the reviewer points out, biochemical analyses are extremely difficult with oocytes due to the limited numbers available and the need to obtain oocytes from live hosts. We therefore used a commercial NAD assay due to its prior validation in mouse oocytes (Wu et al. 2019 Aging Cell) and found that Nampt-depletion significantly reduces NAD levels by around 40% (Fig. 6a).

2. Related to the point above, if the regulation of pre-anaphase and anaphase bipolar spindles is NAD⁺-dependent, the authors should conduct a rescue experiment using the product of the Nampt enzymatic reaction, namely, nicotinamide mononucleotide (NMN). It is important to examine whether NMN can rescue all these defects described in this study. Whether NMN could work or not may be dependent on the type of media (please check the concentration of nicotinamide in the media). If nicotinamide concentration is unphysiologically high (for example, more than 10 microM), NMN may not work. In that case, the authors may want to use nicotinic acid for the rescue experiment.

We are very grateful to the reviewer for suggesting this valuable experiment. Importantly, we found that NMN rescues defects in spindle lengths and PB size in Nampt-depleted oocytes (Fig. 6e, f). Along with evidence using a higher dose of FK866 as well as another highly specific Nampt inhibitor, STF-118804 (Matheny et al. 2013 Chem Biol)(Fig. 6b-d), we present robust evidence in the revised version that reduced NAD levels are in fact an important cause of compromised asymmetry.

3. The authors' FK866 experiment is inconclusive because it is likely that the concentration of FK866 that they used is too low (25 nM). Primary mouse cells tend to require high concentrations of FK866. For example, 500 nM FK866 is necessary to effectively inhibit Nampt in primary hepatocytes. Therefore, the authors should check higher concentrations of FK866 as far as those concentrations are not toxic to mouse oocytes. If the toxicity becomes an issue, the Nampt knockdown is better. In any case, the authors should check NAD⁺ levels in all these experiments.

The reviewer very rightly challenges whether we achieved substantial inhibition of Nampt at the dose of FK866 (25 nM) we used in our initial experiments. As the reviewer points out, 500 nM is required in primary hepatocytes and due to their unique properties, oocytes likely require even higher concentrations. We therefore used 1 μM FK866. Importantly, this level did not impair meiotic maturation but did significantly reduce NAD levels and markedly disrupted asymmetric division (Fig. 6b, c). We further show in the revised version that use of another Nampt inhibitor, STF-11804 (Matheny et al. 2013 Chem Biol), also reduced NAD levels and led to compromised asymmetry (Fig. 6b, d).

4. It is also critical to elaborate what the effector would be in response to NAD⁺ changes. A usual suspect is probably one of the mammalian sirtuin family members, most likely Sirt2 (Please type in "Sirt2 AND spindle formation" to the PubMed. Many related publications can be pulled by these search terms). The authors should examine whether knocking down Sirt2 (or other sirtuins) could possibly cause similar phenotypes in mouse oocytes. If the rescue experiment works, the authors could also examine whether Sirt2 (or other sirtuins) may be required for the rescue. Then, it is important to examine whether tubulin acetylation is changed in mouse oocytes when depleting Sirt2 (or other sirtuins).

We agree entirely that due to its tubulin deacetylation activity, Sirt2 would seem an obvious candidate that could be disrupted when NAD levels are reduced. However, we were surprised to find in new experiments that neither a catalytically inactive Sirt2 construct nor Sirt2 knockdown compromised asymmetry or spindle size (Supplementary Fig. 4). As we discuss in the revised version, we favour a metabolic defect that results in reduced NAD/ATP availability as a key mechanism by which Nampt-depletion compromises spindle function perhaps arising from compromise to other Sirtuin family members such as Sirt1/3 that are critical for mitochondrial activity. We do not, however, rule out other possibilities.

Minor points:

1. Where is the Nampt protein localized in mouse oocytes? Have the authors conducted the immunostaining of Nampt in mouse oocytes?

We agree that this is an important experiment and has also been raised by other reviewers. We have now conducted new immunolocalization experiments and find that Nampt exhibits a striking localisation around the spindle perimeter that coincides with a region of known mitochondrial enrichment (Dalton & Carroll 2013 J Cell Sci)(Fig. 6h-j). Moreover, when Nampt is depleted, both Nampt and mitochondrial localisation around the spindle are compromised (Fig. 6h-l).

2. For the data to which the authors applied two-way ANOVA, the two-way repeated-measures ANOVA would be appropriate.

As requested by the reviewer, we have undertaken analyses using two-way repeated-measures ANOVA instead of two-way ANOVA where appropriate.

3. The Discussion section has many speculations. The authors should address the major points described above, and based on the results, the authors should revise their discussion.

In light of very important new findings, the discussion has been extensively revised.

Reviewers' comments:

Reviewer #1 (Remarks to the Author):

This reviewer acknowledges that the authors have made the best effort to respond to the reviewers' comments. Additional experiments have been performed and the text has been modified accordingly. The initial proposal that the catalytic activity of NAPMT is dispensable for the phenotype associated with NAPMT KD has now been revised. New NAD measurements also strengthen the authors' conclusion that a decrease in NAMPT activity is associated with a decrease in NAD and metabolic instability. Indeed, this decrease in NAD levels may be the underlying defect associated with altered migration of the spindle to the surface and bulging of the membrane. Despite the authors' efforts, no molecular link has been established experimentally to connect the NAPMT activity, NAD levels, and spindle migration. Attempts to implicate Sirtuins or actin cytoskeleton have not panned out. The hypothesis that a decrease in ATP distal to the decrease in NAD is a plausible explanation, but there are no experimental data provided in support of this possibility. Therefore, although intriguing, the findings remain largely correlative.

It is also puzzling that the authors do not attempt to link the spindle phenotype to oocyte aging, where metabolic and mitochondrial disruptions are implicated. It is also puzzling that the senior author in the present paper is coauthor in a manuscript deposited to Biorxiv, where aging, NAD levels, and spindle functions are related. In that study, there is no mention of altered spindle migration or PB morphology, but data suggesting disruption in spindle assembly. There is also mention that altered NAD levels are associated with defective maturation, a finding not reproduced by the Nampt KD. I have not been able to find a plausible explanation for these discrepancies.

Minor points

1. Figures numbers are not included.
2. In Fig3A, the authors compare spindle morphology in oocytes injected with control morpholino and Nampt MO. It is clearly apparent that the length of the spindle in the Nampt morpholino injected oocytes is increased. However, it is also clear that alignment of the chromosome is defective in the Nampt morpholino group, and there is hardly any equatorial alignment visible. Are these images representative of the phenotype? If yes, the authors should discuss potential implications. If not, a better quality oocyte image should be included here.
3. In the same vein in Fig 4g, it appears that larger PB is not the only visible effect of Nampt KD. Blebs are present on the surface of the oocyte which suggest membrane instability. Please comment.
4. In Fig. 7g, She vinculin loading control appears to be decreased with NAMPT MO treatment. A more representative WB should be included.

Reviewer #2 (Remarks to the Author):

Following careful examination of all the reviewer comments, the authors' response and the renewed manuscript, I believe that there is significant improvement supported by mechanistical data and therefore I support the publication of this manuscript in Nature Communications.

One minor comment: In Figure 5, shouldn't "g" be part of "a"?

Reviewer #3 (Remarks to the Author):

In this revised manuscript, the authors adequately addressed all of this reviewer's comments. As a result, the revised manuscript is significantly improved and now ready to be published in Nature Communications.

One minor point that this reviewer want the authors to revise before publication is the sentence in lines 245-246. The authors refer to reference #21, but since this paper, nobody else has ever confirmed the mitochondrial localization of NAMPT. Additionally, the results shown in Fig. 6h-l do not demonstrate the NAMPT localization "within mitochondria." It is interesting that NAMPT "co-localizes" with mitochondria in oocytes, but whether NAMPT is really within mitochondria is something the authors need to address very carefully in the very near future. Accordingly, some sentences in the Discussion section (e.g. line 338) also need to be revised.

We are very grateful for the feedback provided by the reviewers following the submission of our revised manuscript. We note at this stage that Reviewers 2 and 3 are satisfied that the new experiments we provided are sufficient to warrant publication.

Below, we provide a detailed point-by-point response to the Reviewer's comments. Please note that comments from the Reviewers are in italics.

Reviewers' comments:

Reviewer #1 (Remarks to the Author):

This reviewer acknowledges that the authors have made the best effort to respond to the reviewers' comments. Additional experiments have been performed and the text has been modified accordingly. The initial proposal that the catalytic activity of NAPMT is dispensable for the phenotype associated with NAPMT KD has now been revised. New NAD measurements also strengthen the authors' conclusion that a decrease in NAMPT activity is associated with a decrease in NAD and metabolic instability. Indeed, this decrease in NAD levels may be the underlying defect associated with altered migration of the spindle to the surface and bulging of the membrane.

Despite the authors' efforts, no molecular link has been established experimentally to connect the NAPMT activity, NAD levels, and spindle migration. Attempts to implicate Sirtuins or actin cytoskeleton have not panned out. The hypothesis that a decrease in ATP distal to the decrease in NAD is a plausible explanation, but there are no experimental data provided in support of this possibility. Therefore, although intriguing, the findings remain largely correlative.

The Reviewer is now focused on Nampt-related effects “*distal to the decrease in NAD*”. We wish to point out that in the first assessment of our manuscript, this Reviewer's primary question pertained to whether Nampt's role in spindle sizing involved its NAD biosynthetic activity and stated “*The authors do not provide mechanistic data to establish a causative link between the loss of Nampt function and the phenotypes that they describe in considerable detail. Although one could assume that disruption of the NAD salvage pathway is at the root of the phenotype, the experiment suggesting that NAMPT catalytic activity is not required*

rejects this view.” Thus, the major additional mechanistic question posed by this Reviewer pertained to whether Nampt-depletion was producing its effect via reduced NAD levels over and above the finding that Nampt is required for spindle sizing. We have since convincingly shown that this is indeed the case since, firstly, NAD levels are reduced in Nampt-depleted oocytes, secondly, reducing NAD levels using two independent Nampt inhibitors also causes large PBs and importantly, expression of the downstream product of Nampt, NMN, rescues the Nampt-depletion phenotype. **Hence, in addition to showing that Nampt regulates spindle size, we have provided a further level of mechanistic insight by showing that Nampt achieves this, at least in part, by regulating NAD levels.** The Reviewer now seems to be moving the goalpost by asking us to identify yet another mechanistic layer, that is, targets downstream of NAD (+/- ATP). This seems unreasonable after we have convincingly answered the primary query raised by this and the other reviewers in the first revision. Moreover, as we discuss further below, due to the countless regulators and pathways that could potentially be impacted by NAD and/or ATP, pinpointing such targets would be tantamount to searching for a needle in a haystack.

We wish to highlight that we showed that Nampt is part of the Mos-dependent pathway for sizing spindles therefore providing another layer of mechanistic insight. Furthermore, we did try to find a target downstream of NAD that might be affected. As pointed out by Reviewer 3, one of the most likely candidates was Sirt2. However, we found that neither Sirt2-knockdown nor a catalytically inactive Sirt2 mutant impacted asymmetric division. Therefore, we conclude that reduced NAD levels following Nampt-depletion is not impairing asymmetry via an effect on Sirt2 only. We also recently found that loss of Sirt3 only did not affect asymmetric division (Iljas & Homer 2020, *FASEB J*). Added to this, we currently have a paper under review in which we studied oocytes lacking Sirt1 only and again find no impact on asymmetric division. Thus, depleting any one of the three major sirtuins (Sirt1, Sirt2 or Sirt3) on their own does not impact asymmetric division. Significantly, however, our work suggests that lack of effect when a single sirtuin is depleted could be due to compensation by other sirtuins (Iljas & Homer 2020, *FASEB J*). In keeping with this, protein interaction studies point to complementary and redundant roles amongst sirtuins (Yang et al. 2016 *Cell*). This therefore raises the very likely possibility – and one that we favour – that the phenotype is brought about by multiple targets being disrupted simultaneously. The breadth of targets disrupted could be very large indeed since our data show that ATP levels are also affected. We note here that in a separate study in which, NAD and ATP levels were reduced following the depletion of another NAD synthetic enzyme, NMNAT, asymmetrical division was also compromised in mouse oocytes (Wu et al. 2019 *Aging Cell*) although the reasons for this effect were not investigated. It is clearly not feasible to deplete all these myriad NAD and ATP targets in varying permutations to identify the key ones (or specific combinations) responsible for the phenotype we observe. We include a discussion in the revised manuscript surrounding the high likelihood that the phenotype observed is due to defects in multiple targets.

Regarding the Reviewer’s comment *“The hypothesis that a decrease in ATP distal to the decrease in NAD is a plausible explanation, but there are no experimental data provided in support of this possibility”*, we have shown that ATP levels are reduced alongside NAD in Nampt-depleted oocytes. In the revised version, we now provide new data showing that ATP levels are also significantly reduced in oocytes treated with Nampt inhibitors, FK866 and STF-118804 (new Fig. 6h), further strengthening the possibility that reduced ATP may be a contributing factor. We accept that this is not incontrovertible proof that reduced ATP contributes to the phenotype, but it does strengthen the assertion. As we explained above, to

counter the Reviewer's comment that "*there are no experimental data provided in support of this possibility*" would require us to disrupt an endless array of ATP targets either singly or in combination. Even then, disrupting ATP targets might not provide an answer as the phenotype is likely to be brought on by disruptions to both ATP- and NAD-dependent regulators.

The challenge in finding downstream effectors is magnified further by the possibility that at least some of the effects we observe may be spatially restricted to the region of the spindle since we showed that Nampt is spatially enriched in the region of the spindle exactly where mitochondria are also enriched. Moreover, depletion of Nampt impacts peri-spindle mitochondrial enrichment. Another possibility, therefore, which we do not rule out is that Nampt and/or the spindle-proximal mitochondria that are impacted by Nampt-depletion might also exert a scaffolding function over and above a localised metabolic function. There is no straightforward way to formally test these possibilities.

In summary, whilst the other 2 reviewers are satisfied that we have incorporated sufficient mechanistic insight, Reviewer 1 is now asking for further mechanisms downstream of NAD. For the reasons we explained above, elucidating such mechanisms is not straightforward since both NAD and ATP levels are reduced meaning that multiple targets are likely simultaneously impacted added to which, there may well be spatially restricted effects.

It is also puzzling that the authors do not attempt to link the spindle phenotype to oocyte aging, where metabolic and mitochondrial disruptions are implicated. It is also puzzling that the senior author in the present paper is coauthor in a manuscript deposited to Biorxiv, where aging, NAD levels, and spindle functions are related. In that study, there is no mention of altered spindle migration or PB morphology, but data suggesting disruption in spindle assembly. There is also mention that altered NAD levels are associated with defective maturation, a finding not reproduced by the Nampt KD. I have not been able to find a plausible explanation for these discrepancies.

The Reviewer is questioning why our Nampt-depletion phenotype, which we find to be linked with reduced NAD levels and possibly also deranged mitochondrial activity, differs from findings in the aged oocyte model, which is also associated with reduced NAD levels and mitochondrial dysfunction. Of course, it is overly simplistic to equate a phenotype observed when a single protein (in this case Nampt) is acutely depleted *in vitro* in a fully-grown oocyte obtained from a young mouse with the multitude of derangements that undoubtedly accompany ageing and which (importantly) persist *in vivo* throughout the extended growth phase of the oocyte, and which would also impact the surrounding follicular cells during this protracted period (lasting 2-3 weeks in mice and 2-3 months in humans). The consequence is that although NAD levels and mitochondrial function are both abnormal in aged oocytes, there are a multitude of separate abnormalities that accumulate during aging and exert effects throughout the entirety of oocyte growth and follicle development phases. Thus, although both Nampt-depleted oocytes and aged oocytes exhibit reduced NAD levels, a multitude of other abnormalities co-exist in aged oocytes that would not be a feature of fully-grown oocytes from young mice that have been acutely depleted of Nampt.

Minor points

1. Figures numbers are not included.

We request clarification as we are unclear as to what the Reviewer is referring to.

2. In Fig3A, the authors compare spindle morphology in oocytes injected with control morpholino and *Nampt* MO. It is clearly apparent that the length of the spindle in the *Nampt* morpholino injected oocytes is increased. However, it is also clear that alignment of the chromosome is defective in the *Nampt* morpholino group, and there is hardly any equatorial alignment visible. Are these images representative of the phenotype? If yes, the authors should discuss potential implications. If not, a better quality oocyte image should be included here.

We strongly disagree that “*there is hardly any equatorial alignment*” in Fig. 3a. In the Pre-anaphase panels, all chromosomes are clearly congregated at the spindle equator in the *Nampt*MO oocyte with none of them being displaced towards either spindle pole. The region occupied by red fluorescence is very similar in both the *Nampt*MO and control oocytes. Metaphase chromosomes in mouse oocytes are known to undergo oscillatory movements about the equator (see Brunet et al. 1999 J Cell Biol) so the apparent “tightness” of alignment will vary from one timelapse frame to another. We would like to point out Supplementary Fig. 3, for instance, in which chromosome alignment on the particular timelapse frame depicted is very tight for the *Nampt*MO oocyte and seemingly tighter than that for the control oocyte.

3. In the same vein in Fig 4g, it appears that larger PB is not the only visible effect of *Nampt* KD. Blebs are present on the surface of the oocyte which suggest membrane instability. Please comment.

The Reviewer refers to a *Nampt*MO oocyte exhibiting small numbers of blebs. This is by no means a consistent feature specific to *Nampt*-depletion and is also not uncommonly observed in control denuded oocytes during *in vitro* culture. Our analysis of cortical actin levels (Supplementary Fig. 5b, c) does not reveal any objective differences in the cortical cytoskeleton between *Nampt*-depleted and control oocytes that would predispose to altered membrane behaviour.

4. In Fig. 7g, the vinculin loading control appears to be decreased with *NAMPT* MO treatment. A more representative WB should be included.

Although the Reviewer refers to a *NAMPT*MO group in Fig. 7g, the study groups are ControlMO- and MosMO-injected oocytes and there is no *Nampt*MO-injected group. We agree that the vinculin loading does appear to be modestly reduced in the MosMO group. However, in contrast to the vinculin signals, the reduction in *Nampt* levels is stark. Furthermore, we wish to point out that *Nampt* band intensities were analysed by calculating the ratio of the *Nampt* signal to the corresponding vinculin signal intensity thereby correcting for any slight differences in loading. We have nevertheless included a different Western in the revised manuscript and have clarified the details used for analysing blots in the revised Methods.

Reviewer #2 (Remarks to the Author):

Following careful examination of all the reviewer comments, the authors' response and the renewed manuscript, I believe that there is significant improvement supported by mechanistical data and therefore I support the publication of this manuscript in Nature Communications.

We are very pleased that the reviewer now thinks that there is sufficient mechanistical data to support publication.

One minor comment: In Figure 5, shouldn't "g" be part of "a"?

In the revised version of the manuscript we separated parts “a” and “g” as our new data required us to develop the message differently in the main text.

Reviewer #3 (Remarks to the Author):

In this revised manuscript, the authors adequately addressed all of this reviewer's comments. As a result, the revised manuscript is significantly improved and now ready to be published in Nature Communications.

We are very pleased that the reviewer now thinks that the manuscript is significantly improved and now ready to be published.

One minor point that this reviewer want the authors to revise before publication is the sentence in lines 245-246. The authors refer to reference #21, but since this paper, nobody else has ever confirmed the mitochondrial localization of NAMPT. Additionally, the results shown in Fig. 6h-l do not demonstrate the NAMPT localization "within mitochondria." It is interesting that NAMPT "co-localizes" with mitochondria in oocytes, but whether NAMPT is really within mitochondria is something the authors need to address very carefully in the very near future. Accordingly, some sentences in the Discussion section (e.g. line 338) also need to be revised.

We thank the reviewer for raising these points. In the revised version, we allude to the uncertainty surrounding mitochondrial Nampt localisation and include a new reference along these lines (Pittelli et al. 2010 J Biol Chem). We agree as well that co-localisation does not prove that Nampt is within mitochondria, we have modified the revised text accordingly to be more speculative.

REVIEWERS' COMMENTS

Reviewer #1 (Remarks to the Author):

The authors have responded to my concerns and clarified several issues that I have raised. I do not have any further comments.

Reviewer #2 (Remarks to the Author):

After careful evaluation of the original and revised manuscript, the data provided, the reviewers comments and the response of the authors to them, I believe that the manuscript can be published in Nature Communication.

We are very grateful for the feedback provided by the reviewers following the submission of our revised manuscript.

Below, we provide a detailed point-by-point response to the Reviewer's comments. Please note that comments from the Reviewers are in italics.

Reviewer #1 (Remarks to the Author):

The authors have responded to my concerns and clarified several issues that I have raised. I do not have any further comments.

Reviewer #2 (Remarks to the Author):

After careful evaluation of the original and revised manuscript, the data provided, the reviewers comments and the response of the authors to them, I believe that the manuscript can be published in Nature Communication.

We are very pleased with the positive feedback from both Reviewers, neither of which request any further revisions prior to publication.

Yours sincerely

HA Homer MBBS (Honours) MRCOG CCSST FRANZCOG CREI PhD